# Exploration and Anti-Exploration with Distributional Random Network Distillation

## Abstract

Exploration remains a critical issue in deep reinforcement learning for an agent to attain high returns in unknown environments. Although the prevailing exploration Random Network Distillation (RND) algorithm has been demonstrated to be effective in numerous environments, it often needs more discriminative power in bonus allocation. This paper highlights the "bonus inconsistency" issue within RND, pinpointing its primary limitation. To address this issue, we introduce the Distributional RND (DRND), a derivative of the RND. DRND enhances the exploration process by distilling a distribution of random networks and implicitly incorporates pseudo counts to improve the precision of bonus allocation. This refinement encourages agents to engage in more extensive exploration. Our method effectively mitigates the inconsistency issue without introducing significant computational overhead. Both theoretical analysis and experimental results demonstrate the superiority of our approach over the original RND algorithm. Our method excels in challenging online exploration scenarios and effectively serves as an anti-exploration mechanism in D4RL offline tasks.

## 1 Introduction

Exploration is a pivotal consideration in reinforcement learning, especially when dealing with environments that offer sparse or intricate reward information. Several methods have been proposed to promote deep exploration Osband et al. (2016), including count-based and curiosity-driven approaches. Count-based techniques in environments with constrained state spaces rely on recording state visitation frequencies to allocate exploration bonuses (Strehl & Littman (2008); Azar et al. (2017)). However, this method encounters challenges in massive or continuous state spaces. In expansive state spaces, "pseudo counts" have been introduced as an alternative (Bellemare et al. (2016); Lobel et al. (2023); Ostrovski et al. (2017); Machado et al. (2020)). However, establishing a correlation between counts and probability density requires rigorous criteria (Ostrovski et al. (2017)), complicating the implementation of density-based pseudo counts resulting in a significant dependency on network design and hyperparameters.

Curiosity-driven methods motivate agents to explore and learn by leveraging intrinsic motivation. This inherent motivation, often called "curiosity", pushes the agent to explore unfamiliar states or actions. Certain approaches derive intrinsic rewards from the prediction loss of environmental dynamics (Achiam & Sastry (2017); Burda et al. (2018a); Pathak et al. (2017)). As states and actions grow familiar, these methods become more efficient. However, these methods can face difficulties when essential information is missing, or the target function is inherently stochastic, as highlighted by the "noisy-TV" problem (Pathak et al. (2017)). The Random Network Distillation (RND) method uses the matching loss of two networks for a particular state to be the intrinsic motivation (Burda et al. (2018a)). It leverages a randomly initialized target network to generate a fixed value for specific states and trains a prediction network to match this output. RND has demonstrated remarkable results in exploration-demanding environments with sparse rewards, such as Montezuma's Revenge. However, RND has its limitations. While its reliance on network loss for intrinsic rewards lacks a robust mathematical foundation, and its interpretability should be more evident compared to count-based techniques. Moreover, the RND method grapples with the issue of **bonus inconsistency**, which becomes apparent during the initial stages of training when no states have been encountered, leading to bonuses that exhibit considerable deviations from a random distribution. RND struggles to precisely represent the dataset's distribution as training progresses, resulting in indistinguishable bonuses.

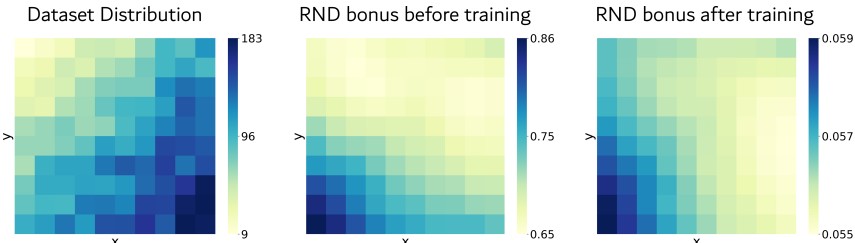

Figure 1: Bonus Heatmap of dataset distribution and RND bonus. The left image illustrates the dataset distribution, the middle image represents the RND bonus before training, and the right image represents the RND bonus after training. A more detailed change process is in Appendix E.8. Ideally, we aim for a uniform bonus distribution before any training and without exposure to the dataset. After extensive training, the expected bonus should inversely correlate with the dataset distribution. The bonus distribution of RND is inconsistent with the desired distribution, indicating a problem with bonus inconsistency. The experiment details can be found in Appendix C

We introduce the Distributional Random Network Distillation (DRND) approach to tackle the challenge of bonus inconsistency in RND. In contrast to the RND method, our approach employs a predictor network to *distill multiple random target networks*. Our findings demonstrate that the DRND predictor effectively operates as a pseudo-count model. This unique characteristic allows DRND to seamlessly merge the advantages of count-based techniques with the RND method, thereby enhancing performance without incurring additional computational and spatial overheads, as the target networks remain fixed and do not require updates. The curiosity-driven RND method and the pseudo-count Coin Flip Network (CFN, Lobel et al. (2023)) method are special cases of our DRND method. Through theoretical analysis and an initial experiment (see Section 5.1), we validate that, compared to RND, DRND demonstrates improved resilience to variations in initial state values, provides a more accurate estimate of state transition frequencies, and better discriminates dataset distributions. As a result, DRND outperforms RND by providing better intrinsic rewards.

In online experiments, we combine the DRND method with Proximal Policy Optimization (PPO, Schulman et al. (2017)). On the image-based exploration benchmark environments Montezuma's Revenge, Gravitar, and Venture, DRND outperform baseline PPO, RND, pseudo-count method CFN, and curiosity-driven method ICM (Pathak et al. (2017)). In continuous control gym-robotics environments, our method also outperforms existing approaches. Furthermore, we demonstrate that DRND can also serve as a good anti-exploration penalty term in the offline setting, confirming its ability to provide a better bonus based on the dataset distribution. We follow the setting of SAC-RND (Nikulin et al. (2023)) and propose a novel offline RL algorithm, SAC-DRND. We run experiments in D4RL (Fu et al. (2020)) offline tasks and find that SAC-DRND outperforms many recent strong baselines across various D4RL locomotion and Antmaze datasets.

## 2    RELATED WORK

**Count-based exploration.** Count-based exploration is a strategy in RL where an agent uses count information to guide its exploration of unknown environments. By keeping track of counts for different states or actions, the agent can estimate the level of unknowns associated with each state or action, prioritizing exploration of those with high unknowns (Bellemare et al. (2016); Machado et al. (2020); Martin et al. (2017); Tang et al. (2017)). These approaches use $r_t = N(s_t)^{-\frac{1}{2}}$ or $r_t = N(s_t, a_t)^{-\frac{1}{2}}$, aiming to balance exploration and exploitation in stochastic MDPs (Strehl & Littman (2008)). Various methods, including CTS (Bellemare et al. (2016)), PixelCNN (Ostrovski et al. (2017)), Successor Counts (Machado et al. (2020)), and CFN (Lobel et al. (2023)), have explored calculating pseudocounts in large state spaces to approximate $N(s_t)$. Furthermore, count-based techniques (Kim & Oh (2023); Hong et al. (2022)) are employed in offline RL to do anti-exploration. While effective in finite state spaces, these methods heavily rely on the network's ability to approximate probability density functions in large state spaces. Accurately estimating density requires a significant number of samples, which limits the effectiveness of counting methods in situations with small sample sizes or regions of low probability density.

**Curiosity-driven exploration.** In curiosity-driven methods, the agent's motivation stems from intrinsic curiosity, often quantified using information-theoretic or novelty-based metrics. One widely used metric involves employing a dynamic model to predict the difference between the expected and actual states, serving as the intrinsic reward (Stadie et al. (2015); Achiam & Sastry (2017); Pathak et al. (2017)), which helps identify unfamiliar patterns, encouraging exploration in less familiar areas. Alternatively, some approaches use information gain as an intrinsic reward (Still & Precup (2012); Houthooft et al. (2016)). Still, they demand computationally intensive network fitting and can struggle in highly stochastic environments due to the "noise TV" problem (Burda et al. (2018b)).

Another curiosity-driven method is RND (Burda et al. (2018b)), which is a prominent RL exploration baseline. RND employs two neural networks: a static prior and a trainable predictor. Both networks map states to embeddings, with state novelty assessed based on their prediction error, which serves as an exploration bonus. This simplicity has bolstered RND's popularity in exploration algorithms and demonstrated its potential in supervised settings, even suggesting its use as an ensemble alternative for estimating epistemic uncertainty (Ciosek et al. (2019); Kuznetsov et al. (2020)). However, common practices, such as using identical architectures for both networks and estimating novelty solely from states, can result in substantial inconsistencies in reward bonuses.

**Anti-Exploration in Model-free Offline RL** Offline RL addresses the problem of learning policies from a logged static dataset. Model-free offline algorithms do not require an estimated model and focus on correcting the extrapolation error (Fujimoto et al. (2019)) in the off-policy algorithms. The first category emphasizes regularizing the learned policy to align with the behavior policy (Kostrikov et al. (2021); Wang et al. (2018; 2020); Wu et al. (2019); Xie et al. (2021); Fujimoto & Gu (2021)). The second category aims to prevent the OOD actions by modifying the value function (Kumar et al. (2020); Lyu et al. (2023; 2022b); An et al. (2021); Ghasemipour et al. (2022); Yang et al. (2022)). These methods employ dual penalization techniques in actor-critic algorithms to facilitate effective offline RL policy learning. These approaches can be further categorized into ensemble-free methods and ensemble-based methods. The ensemble-based methods quantify the uncertainty with ensemble techniques to obtain a robust value function, such as SAC-N (An et al. (2021)) and RORL (Yang et al. (2022)). The ensemble-free methods adapt conservatism to a value function instead of many value functions (Kumar et al. (2020); Lyu et al. (2022b); Rezaeifar et al. (2022)). These methods require punishment for states and actions outside of the dataset distribution, which is called an "anti-exploration" bonus (Rezaeifar et al. (2022)) for the agent. Unlike online RL, where novelty bonuses incentivize exploration, offline RL leans towards conservatism, aiming to reduce rewards in uncharted scenarios. In this work, we introduce a distributional random network distillation approach to serve as a novel anti-exploration method, demonstrating the efficacy of SAC-DRND across various offline RL datasets.

## 3 PRELIMINARIES

**MDP.** We base our framework on the conventional Markov Decision Process (MDP) formulation as described in (Sutton et al. (1998)). In this setting, an agent perceives an observation $o \in \mathcal{O}$ and executes an action $a \in \mathcal{A}$. The transition probability function, denoted by $P(s'|s,a)$, governs the progression from the current state $s$ to the subsequent state $s'$ upon the agent's action $a$. Concurrently, the agent is awarded a reward $r$, determined by the reward function $r : \mathcal{A} \times \mathcal{S} \to \mathbb{R}$. The agent's objective is to ascertain a policy $\pi(a|o)$ that optimizes the anticipated cumulative discounted returns, represented as $\mathbb{E}_\pi \left[ \sum_{t=0}^\infty \gamma^t r\left(s_t, a_t\right) \right]$, where $\gamma \in [0,1)$ serves as the discount factor.

**Intrinsic reward.** To enhance exploration, a common approach involves augmenting the agent's rewards in the environment with intrinsic rewards as a bonus. These intrinsic rewards, denoted as $b_t(s_t, a_t)$, incentivize agents to explore unfamiliar states and take unfamiliar actions. In offline RL, the intrinsic reward is an anti-exploration penalty term to discourage OOD actions. Upon incorporating the intrinsic reward $b(s_t, a_t)$ to the original target of Q value function, the adjusted target can be expressed as follows:

$$y_t = \begin{cases} r_t + \lambda b\left(s_t, a_t\right) + \gamma \max_{a'} Q_{\theta'}\left(s_{t+1}, a'\right), & \text{if online} \\ r_t + \gamma \mathbb{E}_{a' \sim \pi(\cdot|s_{t+1})} \left[ Q_{\theta'}\left(s_{t+1}, a'\right) - \lambda b\left(s_{t+1}, a'\right) \right], & \text{if offline} \end{cases}$$

where $\lambda$ is the scale of the bonus for the update of the value net.

## 4 METHOD

The RND method utilizes two neural networks: a fixed, randomly initialized target network $\hat{f} : \mathcal{O} \to \mathbb{R}^k$, and a predictor network $f : \mathcal{O} \to \mathbb{R}^k$ trained on agent-collected data, where $\mathcal{O}$ is the observation space. In this section, we highlight RND's primary issue and introduce our method, Distributional Random Network Distillation (DRND).

### 4.1 BONUS INCONSISTENCIES IN RANDOM NETWORK DISTILLATION

The RND method faces challenges with bonus inconsistency, which can be categorized into initial and final bonus inconsistencies. The initial bonus inconsistency relates to the uneven distribution of bonuses of states at the beginning of training. Addressing this issue is crucial to preventing significant bonus value disparities among states. Conversely, the final bonus inconsistency arises when the final bonuses do not align with the dataset distribution, making it hard for the agent to effectively distinguish between frequently visited states and those encountered relatively fewer times. This issue becomes particularly pronounced after substantial updates to the predictor network, which hinders the agent's ability to engage in deep exploration. This issue is depicted in Figure 1.

To tackle this, we introduce a method that distills a random distribution, enhancing performance with minimal computational overhead and addressing the bonus inconsistency challenges.

### 4.2 DISTILL THE TARGET NETWORK OF RANDOM DISTRIBUTION

Unlike RND, which only has one target network $\bar{f}(s)$, the DRND algorithm has $N$ target networks $\bar{f}_1(s,a), \bar{f}_2(s,a), ..., \bar{f}_N(s,a)$, which are from a random distribution with randomly initialized parameters and do not participate in training. In DRND, we use $s$ as input in the online setting and $(s,a)$ pair as input in the offline setting. For simplicity, we define $x = (s,a)$ (offline setting) or $x = (s)$ (online setting). For each state-action pair $x$, we construct a variable $c(x)$ which satisfies the distribution:

$$c(x) \sim \begin{array}{c|cccc} X & \bar{f}_1(x) & \bar{f}_2(x) & \dots & \bar{f}_N(x) \\ \hline P & \frac{1}{N} & \frac{1}{N} & \dots & \frac{1}{N} \end{array}$$

For simplicity, we use some symbols to record the moments of the distribution:

$$\mu(x) = \mathbb{E}[X] = \frac{1}{N} \sum_{i=1}^{N} \bar{f}_i(x), \quad B_2(x) = \mathbb{E}[X^2] = \frac{1}{N} \sum_{i=1}^{N} (\bar{f}_i(x))^2. \tag{1}$$

Each time $x$ is occurred, $c(x)$ is sampled from this distribution. We use a predictive network $f_\theta(x)$ to learn the variable $c(x)$, although using a fixed network to learn a random variable is impossible. We use the MSE loss function to force $f_\theta(x)$ to align with $c(x)$ and the loss is

$$L(\theta) = \|f_\theta(x) - c(x)\|_2^2. \tag{2}$$

By minimizing the loss, the optimal $f_*(x)$ when the state-action pair $x$ appears $n$ times is

$$f_*(x) = \frac{1}{n} \sum_{i=1}^{n} c_i(x), \tag{3}$$

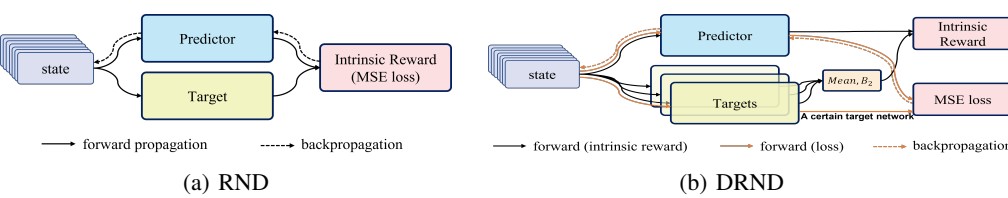

(a) RND                                  (b) DRND

Figure 2: Diagram of RND and DRND. Compared to the RND method that only distills a fixed target network, our method distills a randomly distributed target network and utilizes statistical metrics to assign a bonus to each state.

where $c_i(x)$ is defined as $c(x)$ at the $i$-$th$ occurrence of state $x$. For RND, the more times the predictor network is trained in the same state, the closer the output of the target network is. Therefore, directly using loss as a bonus encourages agents to explore new states. The bonus of our method is not equal to the loss since the loss of random variable fitting is unstable. The expected value of the prediction net is given by

$$\mathbb{E}[f_{\theta^*}(x)] = \mathbb{E}\left[\frac{1}{n}\sum_{i=1}^{n} c_i(x)\right] = \mu(x), \tag{4}$$

where $n$ is the count of occurrences of $x$. After multiple training iterations, this value approaches the mean of the target network distribution. Hence, to measure the deviation from this mean, the first bonus of DRND is defined as

$$b_1(x) = \|f_\theta(x) - \mu(x)\|^2. \tag{5}$$

Compared to predicting the output of one target net, predicting the mean of multiple networks is equivalent to passing through a high pass filter on the output of multiple networks, which can avoid the problem of initial bonus inconsistency due to extreme values in one network. Especially if the network is linear, this bonus inconsistency can be quantitatively calculated.

**Lemma 1.** *Let $\tilde{\theta}$ and $\bar{\theta}_i, i = 1, 2, \ldots, N$ be i.i.d. samples from $p(\theta)$. Given the linear model $f_\theta(x) = \theta^T x$, the expected mean squared error is*

$$E_{\tilde{\theta},\bar{\theta}_1,\bar{\theta}_2,\ldots\bar{\theta}_N}\left[\left\|f_{\tilde{\theta}}(x) - \frac{1}{N}\sum_{i=1}^{N} f_{\bar{\theta}_i}(x)\right\|^2\right] = \left(1 + \frac{1}{N}\right) x^T \Sigma x, \tag{6}$$

*where $\Sigma$ is the variance of $p(\theta)$.*

The complete proof can be seen in Appendix B.1 . Lemma 1 shows that if the predictor parameters and target parameters are sampled from the same distribution, the expectation of the first bonus is a function of input $x$.

**Lemma 2.** *Under the assumptions of Lemma 1, let $x_1, x_2 \in \mathbb{R}^d$, $p(\theta) \sim N(\mu, \sigma^2)$. The bonus difference of $x_1$ and $x_2$ is $\frac{(1+N)\sigma^2}{N}(\|x_2\|^2 - \|x_1\|^2)$.*

*Proof Sketch.* When $p(\theta) \sim N(\mu, \sigma^2)$, the variance of $p(\theta)$ is a constant $\sigma^2$. The right side of Eq. (6) can be rewritten as $\left(1 + \frac{1}{N}\right)\sigma^2\|x\|^2$. So the bonus difference of $x_1$ and $x_2$ is $\left(1 + \frac{1}{N}\right)\sigma^2(\|x_2\|^2 - \|x_1\|^2)$. □

Lemma 2 suggests that when the input $x$ is confined to a bounded interval, and when Eq. (5) is utilized to calculate the initial bonus, the expected maximal difference is modulated by the number of target networks. Importantly, this anticipated discrepancy tends to decrease as $N$ increases. This observation substantiates that our DRND method, equipped with $N$ target networks, exhibits lower bonus inconsistency under a uniform distribution than the RND method, which uses only a single target network.

However, it is essential to note that the network fitting loss determines this bonus. Consequently, it cannot distinguish between states visited multiple times, which stands in contrast to count-based and pseudo-count methods, which do not address the issue of final bonus inconsistency.

### 4.3 THE DRND PREDICTOR IS SECRETLY A PSEUDO-COUNT MODEL

It is essential to track data occurrence frequencies to address inconsistent final bonuses. Traditional count-based methods use large tables to tally state visitations, while pseudo-count strategies use neural networks for estimation, providing a scalable insight into state visits. However, these methods introduce computational and storage complexities, particularly when dealing with high-dimensional inputs. We constructed a statistic that indirectly estimates state occurrences without extra auxiliary functions.

**Lemma 3.** *Let $f_*(x)$ be the optimal function which satisfy Eq. (3), the statistic*

$$y(x) = \frac{[f_*(x)]^2 - [\mu(x)]^2}{B_2(x) - [\mu(x)]^2} \tag{7}$$

*is an unbiased estimator of $1/n$ with consistency.*

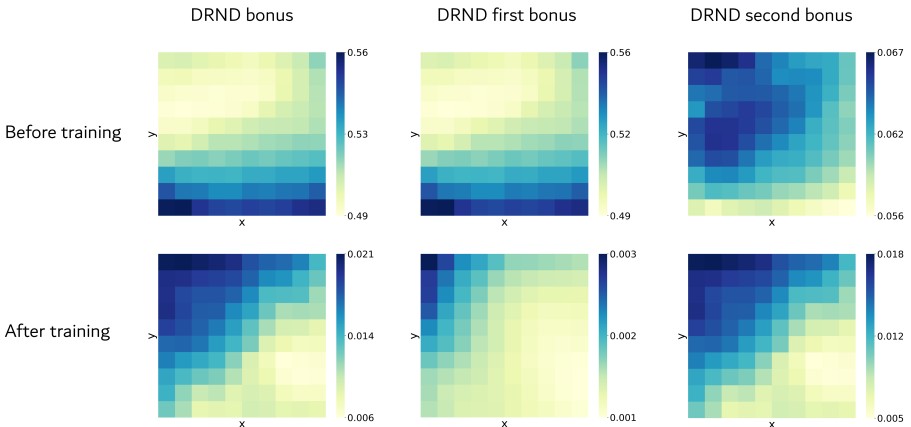

Figure 3: Distribution of DRND bonus. The dataset distribution is the same as Figure 1. These illustrations depict the distribution of the DRND bonus, including the first bonus and the second bonus. The first bonus is predominant before training, and the second bonus becomes more prominent after training.

The complete proof can be found in Appendix B.2. Lemma 3 shows that when $n$ is large, this statistic can effectively recover the number of occurrences of the state $n$, thus implicitly recording the number of occurrences of the state like the pseudo-count method. By minimizing Eq. (2) can make $f_\theta(x)$ and $f_*(x)$ infinitely close, so we replace $f_*(x)$ in $y(x)$ with $f_\theta(x)$ and approximately assume that they are equal. The DRND predictor potentially stores in its weights how much of each state vector is present in the dataset. To correspond to $\sqrt{1/n}$ of the count-based method, the second bonus of the DRND agent is

$$b_2(x) = \sqrt{\frac{[f_\theta(x)]^2 - [\mu(x)]^2}{B_2(x) - [\mu(x)]^2}},\tag{8}$$

which is the estimation of $\sqrt{1/n}$.

### 4.4 BONUS OF THE DRND AGENT

In summary, the total bonus, as seen in Eqs. (5) and (8), is

$$b(s_t, a_t) = \alpha\|f_\theta(x) - \mu(x)\|^2 + (1 - \alpha)\sqrt{\frac{[f_\theta(x)]^2 - [\mu(x)]^2}{B_2(x) - [\mu(x)]^2}}.\tag{9}$$

where $\alpha$ represents the scaling factor for the two bonus terms. Figure 2 shows the diagram of our and RND method. For smaller values of $n$, the variance of the second bonus estimate is substantial, rendering the first bonus a more dependable measure for states with infrequent occurrences. Conversely, as $n$ increases, the variance of the second bonus approaches zero, enhancing its reliability. It's worth noting that during DRND predictor updates, the first bonus diminishes quicker than the second. This dynamic permits a constant $\alpha$ to attain the intended behaviour, as illustrated in Figure 3.

The DRND's target network remains static, and its loss and intrinsic reward calculations do not introduce new backpropagation, keeping computational time similar to the RND algorithm. With $\alpha = 1$ and $N = 1$, the loss and intrinsic reward simplify to $\left\|f_\theta(x) - \bar{f}(x)\right\|^2$, aligning with RND. Unlike count-based and pseudo-count methods, we do not use an extra network or table for state occurrences but estimate using the prediction network's statistics. When $\alpha = 0$ and $c(x) \sim \frac{X \mid -1 \quad 1}{P \mid 0.5 \quad 0.5}$, the loss and intrinsic reward simplify to $\|f_\theta(x)\|^2$ and $\|f_\theta(x)\|$, aligning with the pseudo-count approach CFN.

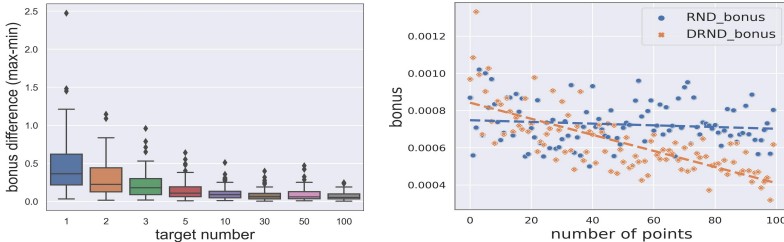

Figure 4: Inconsistency experiments mentioned in Section 5.1. We plot the intrinsic reward distribution of RND and DRND before and after training on a mini-dataset. **Left**: the box plot of the difference between the maximum and minimum intrinsic rewards over 10 independent runs **before** training. **Right**: the intrinsic rewards for each data point **after** training.

# 5 EXPERIMENT

In this section, we provide empirical evaluations of DRND. Initially, we demonstrate that DRND offers a better bonus than RND, both before and after training. Our online experiments reveal that DRND surpasses numerous baselines, achieving the best results in exploration-intensive environments. In the offline setting, we use DRND as an anti-exploration penalty term and propose the SAC-DRND algorithm, which beats strong baselines in many D4RL datasets.

## 5.1 BONUS PREDICTION COMPARISON

In this sub-section, we introduce our inconsistency experiments to compare bonus prediction for both RND and DRND. We created a mini-dataset resembling those used in offline RL or online RL replay buffers in the experiments. This small dataset contains $M$ data categories labeled from 1 to $m$, with each data type occurring $i$ times proportional to its label. Each data point is represented as a one-hot vector with $M$ dimensions, where $M$ is set to 100. We train both the RND and DRND networks on the dataset and record both the initial intrinsic reward and the final intrinsic reward.

The left panel in Figure 4 illustrates the difference in initial intrinsic rewards between RND and our approach, with the x-axis representing the number of target networks. As $N$ increases, the y-axis, representing the range of intrinsic rewards, becomes narrower, resulting in a more uniform reward distribution. In the right panel of Figure 4, we display the intrinsic reward distribution trained on the mini-dataset, showing that DRND's rewards have a stronger correlation with sample count than RND, as indicated by the regression lines.

## 5.2 PERFORMANCE ON ONLINE EXPERIMENTS

Like many other exploration methods, we conduct our DRND approach in Atari games, Adroit environments (Rajeswaran et al. (2017)), and fetch manipulation tasks (Plappert et al. (2018)), which need deep exploration to get a high score. We integrate our method with the PPO (Schulman et al. (2017)) algorithm. We compare our approach with the RND method, the pseudo-count method CFN

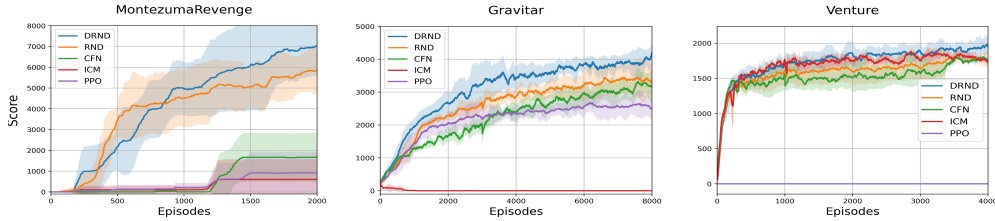

Figure 5: Mean episodic return of DRND method, RND method, and baseline PPO method on three hard exploration Atari games. All curves are averaged over 5 runs.

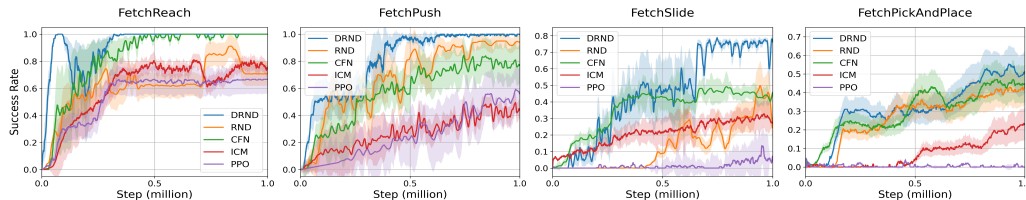

Figure 6: Learning curves in the Adroit continuous control tasks. All curves are averaged over 5 runs.

(Lobel et al. (2023)), the curiosity-driven method ICM (Pathak et al. (2017)), and the baseline PPO method. The solid lines in the figures represent the mean of multiple experiments, and the shading represents the standard deviation interval.

**Atari Games.** We chose three Atari games — Montezuma's Revenge, Gravitar, and Venture — to evaluate our algorithms. These games require deep exploration to achieve high scores, making them ideal for assessing algorithmic exploratory capabilities. We benchmarked our method against the RND and PPO algorithms, with results presented in Figure 5. Our DRND method converges faster and attains the highest final scores in these environments.

**Adroit Experiments.** We further delve into the Adroit continuous control tasks. In these challenges, a robot must skillfully manipulate a hand to perform various actions, such as adjusting a pen's orientation or unlocking a door. Considering the complexity of the tasks and the robot's high-dimensional state space, it becomes imperative to explore methods that can facilitate the robot's learning. Figure 6 illustrates that our DRND method outperforms all the other methods in exploration, especially in the challenging 'Hammer' and 'Relocate' environments. However, in the 'Pen' environment, our method does not exhibit a significant improvement compared to other exploration algorithms. This could be attributed to the relatively simpler nature of this environment, which does not demand deep exploration.

**Fetch Manipulation Tasks.** The Fetch manipulation tasks involve various gym-robotics environments, challenging the Fetch robot arm with complex tasks like reaching, pushing, sliding, and pick-and-place actions. Due to their complexity, these tasks demand advanced exploration strategies. Our evaluation of exploration algorithms in this context highlights their effectiveness in handling intricate robotic manipulations. As shown in Figure 7, our DRND approach excels in assisting the robot in these tasks. Our DRND method effectively combines the strengths of these approaches, outperforming results achievable with either pseudo-count or curiosity-driven methods alone. Consequently, our DRND algorithm performs significantly better than the RND method and other exploration algorithms.

Figure 7: Results on the Fetch manipulation tasks. All curves are averaged over 5 runs.

## 5.3 D4RL OFFLINE EXPERIMENTS

We assessed our method using the D4RL (Fu et al. (2020)) offline datasets, integrating the DRND approach with the SAC algorithm (Haarnoja et al. (2018)). Considering all available datasets in each domain, we tested SAC-DRND on Gym-MuJoCo and the more intricate AntMaze D4RL tasks. Our analysis compares against notable algorithms as detailed in (Rezaeifar et al. (2022)), including IQL (Kostrikov et al. (2021)), CQL (Kumar et al. (2020)), and TD3+BC (Fujimoto & Gu (2021)). It is worth noting that although our method also has $N$ target networks, they are fixed and not trained,

| Dataset | SAC | TD3+BC | CQL | IQL | SAC-RND | ReBRAC | SAC-DRND |
|---|---|---|---|---|---|---|---|
| hopper-random | 9.9 ± 1.5 | 8.5 ± 0.6 | 5.3 ± 0.6 | 10.1 ± 5.9 | 19.6 ± 12.4 | 8.1 ± 2.4 | **32.7** ± 0.4 |
| hopper-medium | 0.8 ± 0.0 | 59.3 ± 4.2 | 61.9 ± 6.4 | 65.2 ± 4.2 | 91.1 ± 10.1 | **102.0** ± 1.0 | 98.5 ± 1.1 |
| hopper-expert | 0.7 ± 0.0 | 107.8 ± 7.0 | 106.5 ± 9.1 | 108.8 ± 3.1 | **109.7** ± 0.5 | 100.1 ± 8.3 | **109.7** ± 0.3 |
| hopper-medium-expert | 0.7 ± 0.0 | 98.0 ± 9.4 | 96.9 ± 15.1 | 85.5 ± 29.7 | **109.8** ± 0.6 | 107.0 ± 6.4 | 108.7 ± 0.5 |
| hopper-medium-replay | 7.4 ± 0.5 | 60.9 ± 18.8 | 86.3 ± 7.3 | 89.6 ± 13.2 | 97.2 ± 9.0 | 98.1 ± 5.3 | **100.5** ± 1.0 |
| hopper-full-replay | 41.1 ± 17.9 | 97.9 ± 17.5 | 101.9 ± 0.6 | 104.4 ± 10.8 | 107.4 ± 0.8 | 107.1 ± 0.4 | **108.2** ± 0.7 |
| halfcheetah-random | 29.7 ± 1.4 | 11.0 ± 1.1 | **31.1** ± 3.5 | 19.5 ± 0.8 | 27.6 ± 2.1 | 29.5 ± 1.5 | 30.4 ± 4.0 |
| halfcheetah-medium | 55.2 ± 27.8 | 48.3 ± 0.3 | 46.9 ± 0.4 | 50.0 ± 0.2 | 66.4 ± 1.4 | 65.6 ± 1.0 | **68.3** ± 0.2 |
| halfcheetah-expert | -0.8 ± 1.8 | 96.7 ± 1.1 | 97.3 ± 1.1 | 95.5 ± 2.1 | 102.6 ± 4.2 | 105.9 ± 1.7 | **106.2** ± 3.7 |
| halfcheetah-medium-expert | 28.4 ± 19.4 | 90.7 ± 4.3 | 95.0 ± 1.4 | 92.7 ± 2.8 | 107.6 ± 2.8 | 101.1 ± 5.2 | **108.5** ± 1.1 |
| halfcheetah-medium-replay | 0.8 ± 1.0 | 44.6 ± 0.5 | 45.3 ± 0.3 | 42.1 ± 3.6 | 51.2 ± 3.2 | 51.0 ± 0.8 | **52.1** ± 4.8 |
| halfcheetah-full-replay | **86.8** ± 1.0 | 75.0 ± 2.5 | 76.9 ± 0.9 | 75.0 ± 0.7 | 81.2 ± 1.3 | 82.1 ± 1.1 | 81.4 ± 1.7 |
| walker2d-random | 0.9 ± 0.8 | 1.6 ± 1.7 | 5.1 ± 1.7 | 11.3 ± 7.0 | 18.7 ± 6.9 | 18.1 ± 4.5 | **21.7** ± 0.1 |
| walker2d-medium | -0.3 ± 0.2 | 83.7 ± 2.1 | 79.5 ± 3.2 | 80.7 ± 3.4 | 91.6 ± 2.8 | 82.5 ± 3.6 | **95.2** ± 0.7 |
| walker2d-expert | 0.7 ± 0.3 | 110.2 ± 0.3 | 109.3 ± 0.1 | 96.9 ± 32.3 | 104.5 ± 22.8 | 112.3 ± 0.2 | **114.0** ± 0.5 |
| walker2d-medium-expert | 1.9 ± 3.9 | 110.1 ± 0.5 | 109.1 ± 0.2 | 112.1 ± 0.5 | 104.6 ± 11.2 | **111.6** ± 0.3 | 109.6 ± 1.0 |
| walker2d-medium-replay | -0.4 ± 0.3 | 81.8 ± 5.5 | 76.8 ± 10.0 | 75.4 ± 9.3 | 88.7 ± 7.7 | 77.3 ± 7.9 | **91.0** ± 2.9 |
| walker2d-full-replay | 27.9 ± 47.3 | 90.3 ± 5.4 | 94.2 ± 1.9 | 97.5 ± 1.4 | 105.3 ± 3.2 | 102.2 ± 1.7 | **109.6** ± 0.7 |
| average score | 16.2 | 67.5 | 73.6 | 72.9 | 82.6 | 81.2 | **86.0** |

| Dataset | SAC | TD3+BC | CQL | IQL | SAC-RND | ReBRAC | SAC-DRND |
|---|---|---|---|---|---|---|---|
| antmaze-umaze | 0.0 | 78.6 | 74.0 | 83.3 ± 4.5 | 97.0 ± 1.5 | **97.8** ± 1.0 | 95.8 ± 2.4 |
| antmaze-umaze-diverse | 0.0 | 71.4 | 84.0 | 70.6 ± 3.7 | 66.0 ± 25.0 | **88.3** ± 13.0 | 87.2 ± 3.2 |
| antmaze-medium-play | 0.0 | 10.6 | 61.2 | 64.6 ± 4.9 | 38.5 ± 29.4 | 84.0 ± 4.2 | **86.2** ± 5.4 |
| antmaze-medium-diverse | 0.0 | 3.0 | 53.7 | 61.7 ± 6.1 | 74.7 ± 10.7 | 76.3 ± 13.5 | **83.0** ±3.8 |
| antmaze-large-play | 0.0 | 0.2 | 15.8 | 42.5 ± 6.5 | 43.9 ± 29.2 | **60.4** ± 26.1 | 53.2 ± 4.1 |
| antmaze-large-diverse | 0.0 | 0.0 | 14.9 | 27.6 ± 7.8 | 45.7 ± 28.5 | **54.4** ± 25.1 | 50.8 ± 10.5 |
| average score | 0.0 | 27.3 | 50.6 | 58.3 | 60.9 | **76.8** | 76.0 |

Table 1: Average normalized scores of ensemble-free algorithms. The figure shows the scores at the final gradient step across 10 different random seeds. We evaluate 10 episodes for MuJoCo tasks and 100 episodes for AntMaze tasks. SAC and TD3+BC scores are taken from (An et al. (2021)). CQL, IQL, SAC-RND, and ReBRAC scores are taken from (Tarasov et al. (2023)). The highest score for each experiment is bolded.

making it ensemble-free. Our SAC-DRND is ensemble-free and only involves training double critics networks. We compare our methods against recent strong model-free offline RL algorithms in Table 1. Additionally, we compare SAC-DRND against strong ensemble-based algorithms like SAC-N in Appendix E.1. Only the results of the ensemble-free methods are shown in the main text. The results are evaluated at the final gradient step over 10 different seeds.

It can be seen that SAC-DRND excels in the majority of MuJoCo tasks, attaining the best results among all ensemble-free methods. On Antmaze tasks, DRND also reached a level similar to SOTA. Compared to SAC-RND, which has comparable computational and storage requirements as our approach, SAC-DRND more effectively captures the dataset distribution, as reflected in its superior average scores and decreased variance. We also conducted experiments on Adroit tasks (Appendix E.2), hyperparameters sensitivity experiments (Appendix E.3) using Expected Online Performance (EOP)(Kurenkov & Kolesnikov (2022)) and offline-to-online experiments (Appendix E.7).

## 6 CONCLUSION

Our research highlights the "bonus inconsistency" issue inherent in RND, which hinders its capacity for deep exploration. We introduce DRND, which distills a random target from a random distribution. Our approach efficiently records state-action occurrences without substantial time and space overhead by utilizing specially designed statistics to extract pseudo-counts. Theoretical analysis and empirical results show our method's effectiveness in tackling bonus inconsistency. We observe promising results across Atari games, gym-robotics tasks, and offline D4RL datasets.

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

# A   DRND PSEUDO-CODE

---

**Algorithm 1** PPO-DRND online pseudo-code

---

**Require:** Number of training steps $M$, number of update steps $K$, number of target networks $N$, scale of intrinsic reward $\lambda$

1: Initialize policy parameters $\phi$
2: Initialize Q-function parameters $\varphi$ and target Q-function parameters $\varphi'$
3: Initialize predictor network parameters $\theta$ and target networks parameters $\theta_1, \theta_2, ..., \theta_N$
4: **for** $i = 1 : N$ **do**
5:      Initialize replay buffer $D$
6:      $d \leftarrow 0, t \leftarrow 0$
7:      $s_0 = $ env.reset()
8:      **while** not $d$ **do**
9:          $a_t \sim \pi(a_t|s_t)$
10:          Rollout $a_t$ and get $(s_{t+1}, r_t, d)$
11:          Compute the mean $\mu(s_t, a_t)$ and second moment $B_2(s_t, a_t)$
12:          Compute intrinsic reward $b(s_{t+1}, a_t)$ using Eq. (9)
13:          Add transition $(s_t, a_t, r_t, b(s_{t+1}, a_t), s_{t+1})$ to $D$
14:          $t \leftarrow t + 1$
15:      **end while**
16:      Normalize the intrinsic rewards contained in $D$
17:      Calculate returns $R_I$ and advantages $A_I$ for intrinsic reward
18:      Calculate returns $R_E$ and advantages $R_E$ for extrinsic reward
19:      Calculate combined advantages $A = R_I + R_E$
20:      $\phi_{old} \leftarrow \phi$
21:      **for** $j = 1 : K$ **do**
22:          Update $\phi$ with gradient ascent using

$$\nabla_\phi \tfrac{1}{|D|} \sum_D \min \left( \tfrac{\pi_\phi(a|s)}{\pi_{\phi_{old}}(a|s)} A, \text{clip} \left( \tfrac{\pi_\phi(a|s)}{\pi_{\phi_{old}}(a|s)}, 1 - \epsilon, 1 + \epsilon \right) A \right)$$

23:          Update $\varphi$ with gradient descent using

$$\nabla_\varphi \tfrac{1}{|D|} \sum_D [Q_\varphi - r_t + \lambda b_\theta(s_t, a_t) + \gamma \max_{a'} Q_{\varphi'}(s_{t+1}, a')]$$

24:          Update $\theta$ using Equation Eq. (2)
25:      **end for**
26:      Update target networks with $\varphi' = (1 - \rho)\varphi' + \rho\varphi$
27: **end for**

---

# B   PROOF

In this section, we will provide all the proofs in the main text.

## B.1   PROOF OF LEMMA 1

$$\mathbb{E}\left[\|f_{\tilde{\theta}}(x) - \frac{1}{N}\sum_{i=1}^{N} f_{\bar{\theta}_i}(x)\|^2\right] = \mathbb{E}\left[\|\tilde{\theta}^T x - \frac{\sum_{i=1}^{N}\bar{\theta}_i^T x}{N}\|^2\right]$$

$$= Var\left(\tilde{\theta}^T x - \frac{\sum_{i=1}^{N}\bar{\theta}_i^T x}{N}\right) - \left(\mathbb{E}\left[\tilde{\theta}^T x - \frac{\sum_{i=1}^{N}\bar{\theta}_i^T x}{N}\right]\right)^2$$

$$= Var\left(\tilde{\theta}^T x - \frac{\sum_{i=1}^{N}\bar{\theta}_i^T x}{N}\right) - \left(\mathbb{E}\left[(\tilde{\theta} - \frac{1}{N}\sum_{i=1}^{N}\bar{\theta}_i)^T x\right]\right)^2.$$

---

**Algorithm 2** SAC-DRND offline pseudo-code

---

**Require:** Number of training steps $M$, number of DRND update steps $K$, number of target networks $N$, scale of intrinsic reward $\lambda_{\text{actor}}, \lambda_{\text{critic}}$, dataset buffer $D$
 1: Initialize policy parameters $\phi$
 2: Initialize two Q-function parameters $\varphi_1, \varphi_2$ and target Q-function parameters $\varphi_1', \varphi_2'$
 3: Initialize predictor network parameters $\theta$ and target networks parameters $\theta_1, \theta_2, ..., \theta_N$
 4: **for** $i = 1 : K$ **do**
 5:     Sample minibatch $(s, a, r, b, s') \sim D$
 6:     Compute the mean $\mu(s_t, a_t)$ and second moment $B_2(s_t, a_t)$
 7:     Update $\theta$ using Equation Eq. (2)
 8: **end for**
 9: **for** $j = 1 : N$ **do**
10:     Sample minibatch $(s, a, r, b, s') \sim D$
11:     Update $\phi$ with gradient ascent using

$$\nabla_\phi \frac{1}{|B|} \sum_B \left[\min_{i=1,2} Q_{\varphi_i}(s, \tilde{a}_\phi(s)) - \beta \log \pi (\tilde{a}_\phi(s) \mid s) - \lambda_{\text{actor}} b_\theta(s, \tilde{a}_\phi(s))\right]$$

        where $\tilde{a}_\phi(s)$ is a sample from $\pi_\phi(.|s)$ by using reparametrization trick
12:     Update each Q-function $Q_{\varphi_i}$ with gradient descent using

$$\nabla_{\varphi_i} \frac{1}{|B|} \sum_B [Q_{\varphi_i} - r_t + \gamma \mathbb{E}_{a' \sim \pi(\cdot|s_{t+1})} [Q_{\varphi_i'}(s_{t+1}, a') - \beta \log \pi_\phi(a'|s') - \lambda_{\text{critic}} b_\theta(s_{t+1}, a')]]$$

        where $a' \sim \pi_\phi(.|s')$
13:     Update target networks with $\varphi_i' = (1 - \rho)\varphi_i' + \rho\varphi_i$
14: **end for**

---

Since $\tilde{\theta}$ and $\bar{\theta}_i$ $(i = 1, 2..., N)$ are i.i.d., $\mathbb{E}\left[(\tilde{\theta} - \frac{1}{N} \sum_{i=1}^N \bar{\theta}_i)^T x\right] = \mathbb{E}[(\tilde{\theta} - \frac{1}{N} \sum_{i=1}^N \bar{\theta}_i)]x = 0$. So we have:

$$\mathbb{E}[\|f_{\tilde{\theta}}(x) - \frac{1}{N} \sum_{i=1}^N f_{\bar{\theta}_i}(x)\|^2]$$

$$= Var\left(\tilde{\theta}^T x - \frac{\sum_{i=1}^N \bar{\theta}_i^T x}{N}\right) - 0$$

$$= Var(\tilde{\theta}^T x) + \frac{1}{N^2} \sum_{i=1}^N Var(\bar{\theta}_i^T x) - \frac{2}{N} \text{Cov}(\tilde{\theta}^T x, \theta_i^T x)$$

$$= Var(\tilde{\theta}^T x) + \frac{1}{N^2} \sum_{i=1}^N Var(\bar{\theta}_i^T x) \quad (\text{Cov}(x, y) = 0 \text{ if } x \text{ and } y \text{ are i.i.d})$$

$$= x^T \Sigma x + \frac{1}{N} x^T \Sigma x$$

$$= (1 + \frac{1}{N}) x^T \Sigma x.$$

### B.2 PROOF PF LEMMA 3

For simplicity, we use some symbols to record the moments of the distribution of $c(x)$

$$\mu(x) = \mathbb{E}[X] = \frac{1}{N} \sum_{i=1}^N \bar{f}_i(x), \qquad B_2(x) = \mathbb{E}[X^2] = \frac{1}{N} \sum_{i=1}^N (\bar{f}_i(x))^2,$$

$$B_3(x) = \mathbb{E}[X^3] = \frac{1}{N} \sum_{i=1}^N (\bar{f}_i(x))^3, \qquad B_4(x) = \mathbb{E}[X^4] = \frac{1}{N} \sum_{i=1}^N (\bar{f}_i(x))^4.$$

The calculation of $f^*(x)$ moment is as follows:

$$\mathbb{E}[f_*(x)] = \mathbb{E}[\frac{1}{n}\sum_{i=1}^{n}c_i(x)] = \frac{1}{n}\mathbb{E}[\sum_{i=1}^{n}c_i(x)] = \mu(x).$$

$$\mathbb{E}[f_*^2(x)] = \mathbb{E}[(\frac{1}{n}\sum_{i=1}^{n}c_i(x))^2]$$

$$= \frac{1}{n^2}\mathbb{E}[(\sum_{i=1}^{n}c_i^2(x) + \sum_{i=1}^{n}\sum_{j\neq i}^{n}c_i(x)c_j(x))]$$

$$= \frac{1}{n^2}\mathbb{E}[nc^2(x) + n(n-1)\mu^2(x)]$$

$$= \frac{B_2(x)}{n} + \frac{n-1}{n}\mu^2(x).$$

$$\mathbb{E}[f_*^4(x)] = \frac{1}{n^4}\mathbb{E}\left[\sum_{i=1}^{n}c_i(x)\right]^4$$

$$= \frac{1}{n^4}\left(\mathbb{E}\left[\sum_{i=1}^{n}c_i(x)^4\right] + 4\mathbb{E}\left[\sum_{i\neq j}c_i^3(x)c_j(x)\right] + 3\mathbb{E}\left[\sum_{i\neq j}c_i^2(x)c_j^2(x)\right]\right.$$

$$\left.+6E\left[\sum_{i\neq j\neq k}c_i(x)c_j(x)c_k^2(x)\right] + \mathbb{E}\left[\sum_{i\neq j\neq k\neq l}c_i(x)c_j(x)c_k(x)c_l(x)\right]\right)$$

$$= \frac{nB_4(x) + 4A_n^2\mu(x)B_3(x) + 3A_n^2B_2^2(x) + 6A_n^3\mu^2(x)B_2(x) + A_n^4\mu^4(x)}{n^4}.$$

$$(A_n^i = \frac{n!}{(n-i)!})$$

The statistic $y(x)$ is:

$$y(x) = \frac{f_*^2(x) - \mu^2(x)}{B_2(x) - \mu^2(x)},$$

and its expectation is:

$$\mathbb{E}[y(x)] = \frac{\mathbb{E}[f_*^2(x)] - \mu^2(x)}{B_2(x) - \mu^2(x)} = \frac{1}{n}.$$

This indicates that the statistic $y(x)$ is an unbiased estimator for the reciprocal of the frequency of $x$. The variance of $y(x)$ is:

$$Var[y(x)] = \frac{Var[f_*^2(x)]}{(B_2(x) - \mu^2(x))^2}$$

$$= \frac{\mathbb{E}[f_*^4(x)] - \mathbb{E}^2[f_*^2(x)]}{(B_2(x) - \mu^2(x))^2}$$

$$= \frac{K_1B_4(x) + K_2\mu(x)B_3(x) + K_3B_2^2(x) + K_4\mu^2(x)B_2(x) + \mu^4(x)}{n^3(B_2(x) - \mu^2(x))^2}$$

where

$$K_1 = 1, \quad K_2 = 4n-4, \quad K_3 = 2n-3,$$
$$K_4 = 4n^2 - 16n + 12, \quad K_5 = -5n^2 + 10n - 6.$$

so we have:

$$\lim_{n\to\infty} Var[y(x)] = 0.$$

When $n$ tends to infinity, the variance of the statistic tends to zero, which reflects the stability or consistency of $y(x)$.

## C  IMPLEMENTATION DETAILS AND EXPERIMENTAL SETTINGS

Our experiments were performed by using the following hardware and software:

- GPUs: NVIDIA GeForce RTX 3090

- Python 3.10.8

- Numpy 1.23.4

- Gymnasium 0.28.1

- Gymnasium-robotics 1.2.2

- Pytorch 1.13.0

- MuJoCo-py 2.1.2.14

- MuJoCo 2.3.1

- D4RL 1.1

- Jax 0.4.13

The architecture of predictor networks in Figure 1 and Figure 3 is 3 linear layers, the input dim is 2, hidden dim and output dim is 16, activate function is ReLU. Target networks' architecture is 2 linear layers, the input dim is 2, hidden dim and output dim is 16, activate function is ReLU.

In our experiments, for a fair comparison, all methods employed the same predictor and target networks. The fundamental parameters of the base algorithm such as learning rate and batch size were kept identical across all methods. For the hyperparameters of the utilized exploration algorithms, we utilized the author-recommended hyperparameters from respective papers (e.g., CFN). Since the CFN method was originally proposed for off-policy strategies, it utilized a trick of importance sampling for sampling from the replay buffer. However, in our PPO-based approach, there is no replay buffer, and we consider the use of the importance sampling trick unfair compared to other methods. Therefore, we only employed the core formula from the paper $b(s) = \sqrt{\frac{1}{d}\|f_\theta(s)\|}$ as the intrinsic reward, where $d$ represents the output dimension of the predictor network.

In our D4RL experiments, all experiments use the dataset of the 'v2' version. We use specific hyperparameters for each task due to varying anti-exploration penalties. Because most of the offline experiment time is spent on gradient calculation of data, we use the faster Jax framework (Bradbury et al. (2018)) than the Pytorch framework (Paszke et al. (2019)) for the experiments. In the online experiments, we still use the easier-to-read and more portable Pytorch framework instead of the faster computing Jax framework because most of the online experiment time is spent interacting with the environment rather than gradient computing.

We employ the 'NoFrameskip-v4' version in our Atari game experiments to execute the environments. These experiments encompass 128 parallel environments and adhere to the default configurations and network architecture as delineated in (Burda et al. (2018b)). For Adroit and Fetch manipulation tasks, we employ the 'v0' version for Adroit tasks and the 'v2' version for Fetch tasks. In the 'Relocate' task, we truncate the episode when the ball leaves the table. These tasks pose a significant challenge for conventional methods to learn from, primarily due to the dataset consisting of limited human demonstrations in a sparse-reward, complex, high-dimensional robotic manipulation task (Lyu et al. (2022a)). We do not include random state restarts, as they may undermine the necessity for exploration by the observations made by (Lobel et al. (2022)). To set the goal locations for the non-default versions of the tasks, we follow the setting of (Lobel et al. (2023)).

In the context of the D4RL framework, we make specific architectural choices. Instead of simply concatenating the state and action dimensions, we employ a bilinear structure in the first layer, as proposed by (Jayakumar et al. (2020)). Additionally, we apply FiLM (Feature-wise Linear Modulation) architecture on the penultimate layer before the nonlinearity. This modification is effective for offline tasks, as indicated by (Nikulin et al. (2023)).

# D  HYPERPARAMETERS

The hyperparameters are shown in Table 4 in online experiments. We employ distinct parameters and networks for Atari games and continuous control environments because Atari game observations are images, while observations for Adroit and Fetch tasks consist of states. The hyperparameters we use in the D4RL offline experiment are shown in Table 2. In D4RL offline datasets, we apply varying scales in each experiment due to the differing dataset qualities, as illustrated in Table 3.

Table 2: Hyperparameters of D4RL offline experiments

| Name | Description | Value |
|------|-------------|-------|
| $lr_{\text{actor}}$ | learning rate of the actor network | 1e-3 (1e-4 on Antmaze) |
| $lr_{\text{critic}}$ | learning rate of the critic network | 1e-3 (1e-4 on Antmaze) |
| $lr_{\text{drnd}}$ | learning rate of the DRND network | 1e-6 (1e-5 on Antmaze) |
| optimizer | type of optimizer | Adam |
| target entropy | target entropy of the actor | -action_dim |
| $\tau$ | soft update rate | 0.005 |
| $\gamma$ | discount return | 0.99 (0.999 on Antmaze) |
| $bs$ | batch size of the dataset | 1024 |
| $h$ | number of hidden layer dimensions | 256 |
| $e$ | number of DRND output dimensions | 32 |
| $n$ | number of hidden layers | 4 |
| $f$ | activation function | ReLU |
| $K$ | number of DRND training epochs | 100 |
| $M$ | maximum iteration number of SAC | 3000 |
| $I$ | gradient updates per iteration | 1000 |
| $N$ | number of DRND target networks | 10 |
| $\alpha$ | the scale of two intrinsic reward items | 0.9 |

Table 3: Anti-exploration scale of D4RL offline datasets

| Dataset Name | $\lambda_{\text{actor}}$ | $\lambda_{\text{critic}}$ |
|--------------|------|------|
| hopper-random | 1.0 | 1.0 |
| hopper-medium | 15.0 | 15.0 |
| hopper-expert | 10.0 | 10.0 |
| hopper-medium-expert | 10.0 | 10.0 |
| hopper-medium-replay | 5.0 | 10.0 |
| hopper-full-replay | 2.0 | 2.0 |
| halfcheetah-random | 0.05 | 0.05 |
| halfcheetah-medium | 1.0 | 0.1 |
| halfcheetah-expert | 5.0 | 5.0 |
| halfcheetah-medium-expert | 0.1 | 0.1 |
| halfcheetah-medium-replay | 0.1 | 0.1 |
| halfcheetah-full-replay | 1.0 | 1.0 |
| walker2d-random | 1.0 | 1.0 |
| walker2d-medium | 10.0 | 10.0 |
| walker2d-expert | 5.0 | 5.0 |
| walker2d-medium-expert | 15.0 | 20.0 |
| walker2d-medium-replay | 5.0 | 10.0 |
| walker2d-full-replay | 3.0 | 3.0 |
| antmaze-umaze | 5.0 | 0.001 |
| antmaze-umaze-diverse | 3.0 | 0.001 |
| antmaze-medium-play | 3.0 | 0.005 |
| antmaze-medium-diverse | 2.0 | 0.001 |
| antmaze-large-play | 1.0 | 0.01 |
| antmaze-large-diverse | 0.5 | 0.01 |

Table 4: Hyperparameters of online experiments

| Name | Description | Value |
|------|-------------|-------|
| $lr_{\text{actor}}$ | learning rate of the actor network | 3e-4 (1e-4 on Atari) |
| $lr_{\text{critic}}$ | learning rate of the critic network | 3e-4 (1e-4 on Atari) |
| $lr_{\text{drnd}}$ | learning rate of the DRND network | 3e-4 (1e-4 on Atari) |
| optimizer | type of optimizer | Adam |
| $\tau$ | soft update rate | 0.005 |
| $\gamma$ | discount return | 0.99 |
| $\lambda_{\text{GAE}}$ | coefficient of GAE | 0.95 |
| $\epsilon$ | PPO clip coefficient | 0.1 |
| $M$ | number of environments | 128 |
| $h$ | number of hidden layer dimensions | 64 (512 on Atari) |
| $e$ | number of output dimensions | 64 (512 on Atari) |
| $f$ | activation function | ReLU |
| $K$ | number of training epochs | 4 |
| $N$ | number of DRND target networks | 10 |
| $\lambda$ | coefficient of intrinsic reward | 1 |
| $\alpha$ | the scale of two intrinsic reward items | 0.9 |

## E    ADDITIONAL EXPERIMENTAL RESULTS

### E.1    COMPARING TO ENSEMBLE-BASED METHODS

As described in (Osband et al. (2016)), the ensemble method estimates the Q-posterior, leading to varied predictions and imposing significant penalties in regions with limited data. We add the results of ensemble-based methods like SAC-N (An et al. (2021)), EDAC (An et al. (2021)), and RORL (Yang et al. (2022)). Table 5 displays our results in these experiments. An underlined number represents the peak value for ensemble-free methods, while a **bold** number denotes each task's top score. SAC-DRND outperforms most ensemble-based methods, such as SAC-N and RORL, in total scores on most MuJoCo tasks. For Antmaze tasks, our method leads among ensemble-free approaches and holds its own against ensemble-based methods.

### E.2    RESULTS ON ADROIT TASKS

In this subsection, we show the scores of SAC-DRND on Adroit tasks in Table 6.

### E.3    EXPECTED ONLINE PERFORMANCE

We calculated the EOP on Gym-MuJoCo and AntMaze tasks, as shown in Table 7.

We also show the EOP line for each task, as shown in Figure 8.

### E.4    PARAMETER STUDY ON THE NUMBER OF TARGET NETWORK

Our study explored the relationship between the number of different targets and their corresponding final scores in both online MuJoCo tasks and D4RL offline tasks. In our approach, if $\alpha$ is not equal to 1, then $N$ must satisfy the condition $N > 1$. In the following charts, we fill in the values of RND at $N = 1$ as a reference for the single target network results.

### E.4.1    ONLINE TASKS

We conduct the adversarial attack experiments with different numbers of target networks in DRND. As shown in Figure 9, the robustness of DRND generally improves with an increase in the target number $N$. Considering both runtime and performance, we chose $N = 10$ as the optimal number of targets for our online experiments.

| Dataset | SAC | TD3+BC | CQL | IQL | SAC-RND | ReBRAC | SAC-N | EDAC | RORL | SAC-DRND |
|---|---|---|---|---|---|---|---|---|---|---|
| | | | | Ensemble-free | | | Ensemble-based | | | Ours |
| hopper-random | 9.9 ± 1.5 | 8.5 ± 0.6 | 5.3 ± 0.6 | 10.1 ± 5.9 | 19.6 ± 12.4 | 8.1 ± 2.4 | 28.0 ± 0.9 | 25.3 ± 10.4 | 31.4 ± 0.1 | **32.7 ± 0.4** |
| hopper-medium | 0.8 ± 0.0 | 59.3 ± 4.2 | 61.9 ± 6.4 | 65.2 ± 4.2 | 91.1 ± 10.1 | 102.0 ± 1.0 | 100.3 ± 0.3 | 101.6 ± 0.6 | **104.8 ± 0.1** | 98.5 ± 1.1 |
| hopper-expert | 0.7 ± 0.0 | 107.8 ± 7.0 | 106.5 ± 9.1 | 108.8 ± 3.1 | 109.7 ± 0.5 | 100.1 ± 8.3 | 110.3 ± 0.3 | 110.1 ± 0.1 | **112.8 ± 0.2** | 109.7 ± 0.3 |
| hopper-medium-expert | 0.7 ± 0.0 | 98.0 ± 9.4 | 96.9 ± 15.1 | 85.5 ± 29.7 | 109.8 ± 0.6 | 107.0 ± 6.4 | 110.1 ± 0.3 | 110.7 ± 0.1 | **112.7 ± 0.2** | 108.7 ± 0.5 |
| hopper-medium-replay | 7.4 ± 0.5 | 60.9 ± 18.8 | 86.3 ± 7.3 | 89.6 ± 13.2 | 97.2 ± 9.0 | 98.1 ± 5.3 | 101.8 ± 0.5 | 101.0 ± 0.5 | **102.8 ± 0.5** | 100.5 ± 1.0 |
| hopper-full-replay | 41.1 ± 17.9 | 97.9 ± 17.5 | 101.9 ± 0.6 | 104.4 ± 10.8 | 107.4 ± 0.8 | 107.1 ± 0.4 | 102.9 ± 0.3 | 105.4 ± 0.7 | - | **108.2 ± 0.7** |
| halfcheetah-random | 29.7 ± 1.4 | 11.0 ± 1.1 | **31.1 ± 3.5** | 19.5 ± 0.8 | 27.6 ± 2.1 | 29.5 ± 1.5 | 28.0 ± 0.9 | 28.4 ± 1.0 | 28.5 ± 0.8 | 30.4 ± 4.0 |
| halfcheetah-medium | 55.2 ± 27.8 | 48.3 ± 0.3 | 46.9 ± 0.4 | 50.0 ± 0.2 | 66.4 ± 1.4 | 65.6 ± 1.0 | 67.5 ± 1.2 | 65.9 ± 0.6 | 66.8 ± 0.7 | **68.3 ± 0.2** |
| halfcheetah-expert | -0.8 ± 1.8 | 96.7 ± 1.1 | 97.3 ± 1.1 | 95.5 ± 2.1 | 102.6 ± 4.2 | 105.9 ± 1.7 | 105.2 ± 2.6 | **106.8 ± 3.4** | 105.2 ± 0.7 | 106.2 ± 3.7 |
| halfcheetah-medium-expert | 28.4 ± 19.4 | 90.7 ± 4.3 | 95.0 ± 1.4 | 92.7 ± 2.8 | 107.6 ± 2.8 | 101.1 ± 5.2 | 107.1 ± 2.0 | 106.3 ± 1.9 | 107.8 ± 1.1 | **108.5 ± 1.1** |
| halfcheetah-medium-replay | 0.8 ± 1.0 | 44.6 ± 0.5 | 45.3 ± 0.3 | 42.1 ± 3.6 | 51.2 ± 3.2 | 51.0 ± 0.8 | **63.9 ± 0.8** | 61.3 ± 1.9 | 61.9 ± 1.5 | 52.1 ± 4.8 |
| halfcheetah-full-replay | **86.8 ± 1.0** | 75.0 ± 2.5 | 76.9 ± 0.9 | 75.0 ± 0.7 | 81.2 ± 1.3 | 82.1 ± 1.1 | 84.5 ± 1.2 | 84.6 ± 0.9 | - | 81.4 ± 1.7 |
| walker2d-random | 0.9 ± 0.8 | 1.6 ± 1.7 | 5.1 ± 1.7 | 11.3 ± 7.0 | 18.7 ± 6.9 | 18.1 ± 4.5 | **21.7 ± 0.0** | 16.6 ± 7.0 | 21.4 ± 0.2 | **21.7 ± 0.1** |
| walker2d-medium | -0.3 ± 0.2 | 83.7 ± 2.1 | 79.5 ± 3.2 | 80.7 ± 3.4 | 91.6 ± 2.8 | 82.5 ± 3.6 | 87.9 ± 0.2 | 92.5 ± 0.8 | **102.4 ± 1.4** | 95.2 ± 0.7 |
| walker2d-expert | 0.7 ± 0.3 | 110.2 ± 0.3 | 109.3 ± 0.1 | 96.9 ± 32.3 | 104.5 ± 22.8 | 112.3 ± 0.2 | 107.4 ± 2.4 | 115.1 ± 1.9 | **115.4 ± 0.5** | 114.0 ± 0.5 |
| walker2d-medium-expert | 1.9 ± 3.9 | 110.1 ± 0.5 | 109.1 ± 0.2 | 112.1 ± 0.5 | 104.6 ± 11.2 | 111.6 ± 0.3 | 116.7 ± 0.4 | 114.7 ± 0.9 | **121.2 ± 1.5** | 109.6 ± 1.0 |
| walker2d-medium-replay | -0.4 ± 0.3 | 81.8 ± 5.5 | 76.8 ± 10.0 | 75.4 ± 9.3 | 88.7 ± 7.7 | 77.3 ± 7.9 | 78.7 ± 0.7 | 87.1 ± 2.4 | 90.4 ± 0.5 | **91.0 ± 2.9** |
| walker2d-full-replay | 27.9 ± 47.3 | 90.3 ± 5.4 | 94.2 ± 1.9 | 97.5 ± 1.4 | 105.3 ± 3.2 | 102.2 ± 1.7 | 94.6 ± 0.5 | 99.8 ± 0.7 | - | **109.6 ± 0.7** |
| average score | 16.2 | 67.5 | 73.6 | 72.9 | 82.6 | 81.2 | 84.4 | 85.2 | 85.7 | **86.0** |

| Dataset | SAC | TD3+BC | CQL | IQL | SAC-RND | ReBRAC | RORL | MSG | SAC-DRND |
|---|---|---|---|---|---|---|---|---|---|
| antmaze-umaze | 0.0 | 78.6 | 74.0 | 83.3 ± 4.5 | 97.0 ± 1.5 | 97.8 ± 1.0 | 97.7 ± 1.9 | **97.9 ± 1.3** | 95.8 ± 2.4 |
| antmaze-umaze-diverse | 0.0 | 71.4 | 84.0 | 70.6 ± 3.7 | 66.0 ± 25.0 | 88.3 ± 13.0 | **90.7 ± 2.9** | 79.3 ± 3.0 | 87.2 ± 3.2 |
| antmaze-medium-play | 0.0 | 10.6 | 61.2 | 64.6 ± 4.9 | 38.5 ± 29.4 | 84.0 ± 4.2 | 76.3 ± 2.5 | 85.9 ± 3.9 | **86.2 ± 5.4** |
| antmaze-medium-diverse | 0.0 | 3.0 | 53.7 | 61.7 ± 6.1 | 74.7 ± 10.7 | 76.3 ± 13.5 | 69.3 ± 3.3 | **84.6 ± 5.2** | 83.0 ± 3.8 |
| antmaze-large-play | 0.0 | 0.2 | 15.8 | 42.5 ± 6.5 | 43.9 ± 29.2 | 60.4 ± 26.1 | 16.3 ± 11.1 | **64.3 ± 12.7** | 53.2 ± 4.1 |
| antmaze-large-diverse | 0.0 | 0.0 | 14.9 | 27.6 ± 7.8 | 45.7 ± 28.5 | 54.4 ± 25.1 | 41.0 ± 10.7 | **71.3 ± 5.3** | 50.8 ± 10.5 |
| average score | 0.0 | 27.3 | 50.6 | 58.3 | 60.9 | 76.8 | 65.2 | **80.5** | 76.0 |

Table 5: Average normalized scores of algorithms. The figure shows the scores of MuJoCo's final model evaluated 10 times (AntMaze 100 times) on training seeds and 10 random seeds. SAC, SAC-N, and EDAC scores are taken from (An et al. (2021)). CQL, IQL, SAC-RND, and ReBRAC scores are taken from (Tarasov et al. (2023)). RORL scores are taken from (Yang et al. (2022)). MSG scores are taken from (Ghasemipour et al. (2022)).

| Task Name | BC | TD3+BC | IQL | CQL | SAC-RND | ReBRAC | SAC-DRND |
|---|---|---|---|---|---|---|---|
| pen-human | 34.4 | 81.8 ± 14.9 | 81.5 ± 17.5 | 37.5 | 5.6 ± 5.8 | **103.5 ± 14.1** | 42.3 ± 11.8 |
| pen-cloned | 56.9 | 61.4 ± 19.3 | 77.2 ± 17.7 | 39.2 | 2.5 ± 6.1 | **91.8 ± 21.7** | 39.5 ± 33.4 |
| pen-expert | 85.1 | 146.0 ± 7.3 | 133.6 ± 16.0 | 107.0 | 45.4 ± 22.9 | **154.1 ± 5.4** | 65.0 ± 17.1 |
| door-human | 0.5 | -0.1 ± 0.0 | 3.1 ± 2.0 | **9.9** | 0.0 ± 0.1 | 0.0 ± 0.0 | 1.3 ± 0.8 |
| door-cloned | -0.1 | 0.1 ± 0.6 | 0.8 ± 1.0 | 0.4 | 0.2 ± 0.8 | **1.1** ± 2.6 | 0.3 ± 0.1 |
| door-expert | 34.9 | 84.6 ± 44.5 | **105.3 ± 2.8** | 101.5 | 73.6 ± 26.7 | 104.6 ± 2.4 | 85.3 ± 37.9 |
| hammer-human | 1.5 | 0.4 ± 0.4 | 2.5 ± 1.9 | **4.4** | -0.1 ± 0.1 | 0.2 ± 0.2 | 0.3 ± 0.2 |
| hammer-cloned | 0.8 | 0.8 ± 0.7 | 1.1 ± 0.5 | 2.1 | 0.1 ± 0.4 | **6.7** ± 3.7 | 1.1 ± 0.8 |
| hammer-expert | 125.6 | 117.0 ± 30.9 | 129.6 ± 0.5 | 86.7 | 24.8 ± 39.4 | **133.8** ± 0.7 | 37.1 ± 47.2 |
| relocate-human | 0.0 | -0.2 ± 0.0 | 0.1 ± 0.1 | **0.2** | 0.0 ± 0.0 | 0.0 ± 0.0 | 0.0 ± 0.1 |
| relocate-cloned | -0.1 | -0.1 ± 0.1 | 0.2 ± 0.4 | -0.1 | 0.0 ± 0.0 | **0.9** ± 1.6 | 0.0 ± 0.0 |
| relocate-expert | 101.3 | **107.3** ± 1.6 | 106.5 ± 2.5 | 95.0 | 3.4 ± 4.5 | 106.6 ± 3.2 | 10.1 ± 7.1 |
| Average w/o expert | 11.7 | 18.0 | 20.8 | 11.7 | 1.0 | **25.5** | 10.6 |
| Average | 36.7 | 49.9 | 53.4 | 40.3 | 12.9 | **58.6** | 23.5 |

Table 6: Average normalized scores on Adroit tasks. There is still a significant improvement compared to SAC-RND, from 12.9 to 23.5. This illustrates the superiority of DRND compared to RND. In addition, the average score without using the expert dataset has also improved significantly, reaching a level comparable to CQL(11.7), which benefits from the performance in the Pen environment.

## E.4.2 OFFLINE TASKS

The results are shown in Table 8. The results indicate that the average score demonstrates an upward trend as the number of targets increases. At the same time, its variance decreases, which suggests that a higher number of targets generally leads to improved and more consistent outcomes. However,

| Domain | Algorithm | 1 policy | 2 policies | 3 policies | 5 policies | 10 policies | 15 policies | 20 policies |
|--------|-----------|----------|-----------|-----------|-----------|-------------|-------------|-------------|
| Gym-MuJoCo | **TD3+BC** | 49.8 ± 21.4 | 61.0 ± 14.5 | 65.3 ± 9.3 | 67.8 ± 3.9 | - | - | - |
| | **IQL** | 65.0 ± 9.1 | 69.9 ± 5.6 | 71.7 ± 3.5 | 72.9 ± 1.7 | 73.6 ± 0.8 | 73.8 ± 0.7 | 74.0 ± 0.6 |
| | **ReBRAC** | 62.0 ± 17.1 | 70.6 ± 9.9 | 73.3 ± 5.5 | 74.8 ± 2.1 | 75.6 ± 0.8 | 75.8 ± 0.6 | 76.0 ± 0.5 |
| | **SAC-DRND** | **69.9** ± 30.1 | **73.2** ± 19.0 | **79.4** ± 11.9 | **82.5** ± 7.8 | **84.0** ± 6.0 | **84.9** ± 3.1 | **85.3** ± 2.0 |
| AntMaze | **TD3+BC** | 6.9 ± 7.0 | 10.7 ± 6.8 | 13.0 ± 6.0 | 15.5 ± 4.6 | - | - | - |
| | **IQL** | 29.8 ± 15.5 | 38.0 ± 15.4 | 43.1 ± 13.8 | 48.7 ± 10.2 | 53.2 ± 4.4 | 54.3 ± 2.1 | 54.7 ± 1.2 |
| | **ReBRAC** | 67.9 ± 10.0 | 73.6 ± 7.4 | 76.1 ± 5.5 | 78.3 ± 3.4 | 79.9 ± 1.7 | 80.4 ± 1.1 | - |
| | **SAC-DRND** | **69.3** ± 15.9 | **75.3** ± 10.1 | **78.5** ± 7.6 | **81.5** ± 4.0 | **83.7** ± 3.1 | **84.5** ± 1.5 | **84.9** ± 0.9 |

Table 7: Expected Online Performance on Gym-MuJoCo and AntMaze tasks. We calculate the mean value of different domains like the way in ReBRAC. The results show SAC-DRND has the best performance.

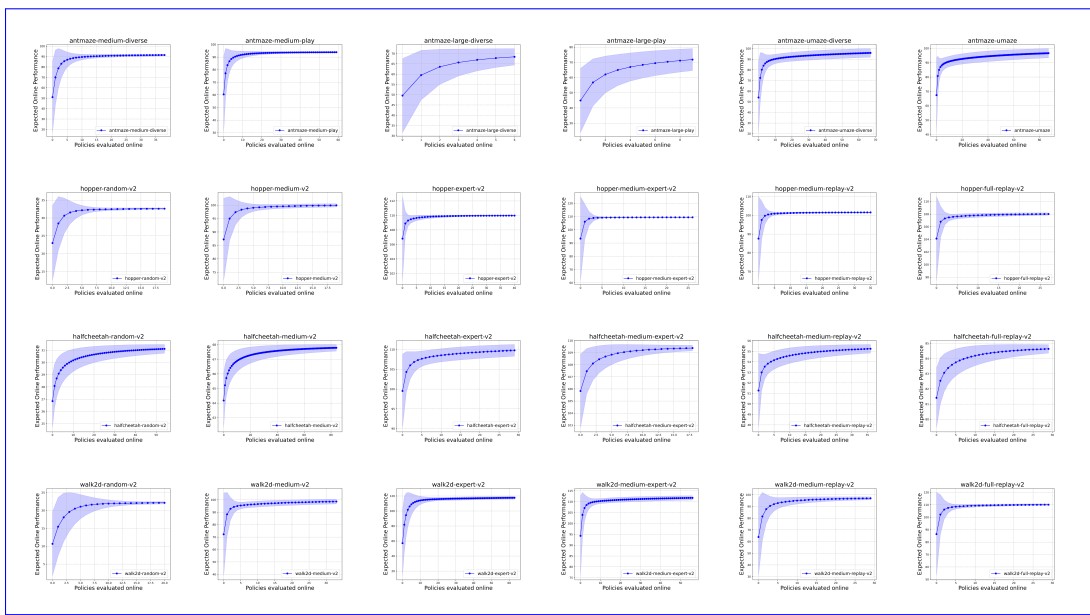

Figure 8: Expected Online Performance lines for Gym-MuJoCo and AntMaze.

it's worth noting that there are diminishing returns; for instance, the differences between the results at $N = 10$ and $N = 20$ are marginal. Considering these considerations, we chose $N = 10$ for our offline experiments. Furthermore, the algorithm exhibits limited sensitivity to variations in the number of targets in online and offline settings.

Table 8: Parameter study of $N$ in offline tasks

| Dataset \ N | 1 | 3 | 5 | 10 | 20 |
|-------------|---|---|---|-----|-----|
| hopper-medium | 92.1 ± 8.4 | 93.3 ± 3.7 | 97.8 ± 2.4 | 98.5 ± 1.1 | **99.0** ± 0.6 |
| halfcheetah-medium | 66.4 ± 1.4 | 65.8 ± 1.8 | 66.7 ± 0.6 | 67.3 ± 0.2 | **67.4** ± 0.4 |
| walker2d-medium | 91.6 ± 2.8 | 94.5 ± 0.9 | 94.0 ± 1.6 | **95.2** ± 1.2 | 94.7 ± 1.2 |
| average score | 83.4 | 84.5 | 86.2 | **87.0** | **87.0** |

### E.5 RUNTIME COMPARISON

To verify no significant increase in computational overhead between our method and the RND method, we conducted experiments on the medium datasets in the offline D4RL tasks, comparing the

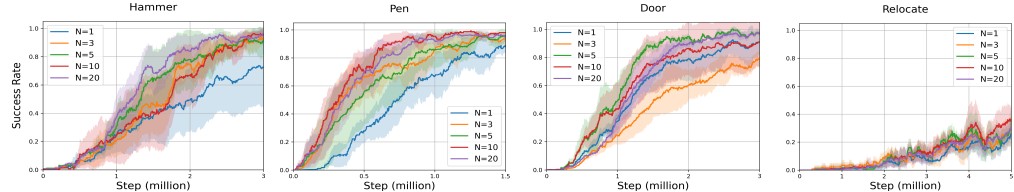

Figure 9: Training curves with different $N$ in Adroit tasks. All curves are averaged over 5 runs.

computational costs of both methods, as shown in Figure 10. It can be observed that the runtime of our method is slightly less than that of the RND method. And it can be seen that as the number of targets increases, the running time does not significantly improve.

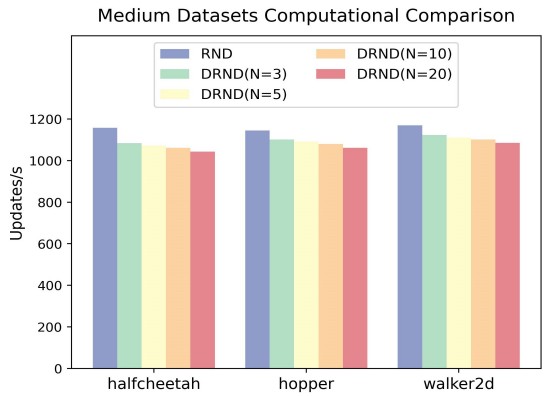

Figure 10: Comparison of updates per second between the RND and DRND methods. We assessed the execution time on a GPU (RTX 3090 24G) and one CPU (Intel(R) Xeon(R) Gold 6226R CPU) over 1M standard updates, using a batch size of 256 with the same network structure.

### E.6 PARAMETER STUDY ON $\alpha$

In this subsection, we provide the results of different $\alpha$ with both online and offline tasks. We use varying $\alpha \in \{0, 0.1, 0.5, 0.9, 1\}$.

### E.6.1 ONLINE TASKS

We study the performance under attacks with different $\alpha$ in online tasks. We chose Adroit continuous control environments as our experiment environments. In the results shown in Figure 11, We observed that the performance is excellent when $\alpha = 0.5$ or $\alpha = 0.9$ in all four environments. The performance when $\alpha = 1$ is not as good as when $\alpha = 0.9$, which indirectly confirms the effect of the second bonus term. We chose $\alpha = 0.9$ as the hyperparameter for our online experiments.

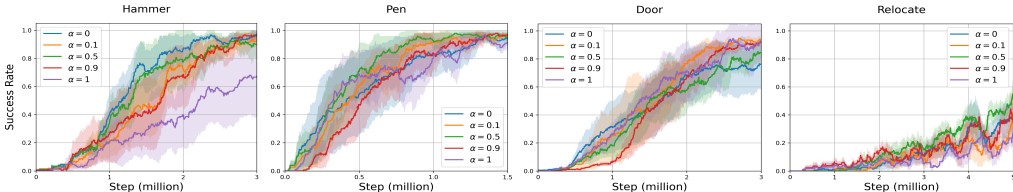

Figure 11: Training curves with different $\alpha$ in Adroit tasks. All curves are averaged over 5 runs.

### E.6.2 OFFLINE TASKS

We examine the influence of $\alpha$ on offline tasks using the D4RL dataset. We employ various values of $\alpha$ to train an offline agent on the 'medium' datasets. The final scores are presented in Table 9, and the training curves are shown in Figure 12. It is observed that in some cases, when $\alpha = 0.9$, the final score is higher, and the training curve exhibits greater stability. Consequently, we consistently opted for $\alpha = 0.9$ in our offline experiments. When $\alpha = 1$, only the first bonus term comes into play, and the results are not as favorable as when $\alpha = 0.9$, demonstrating the effectiveness of the second bonus term. Additionally, when examining the final results, it becomes evident that our first bonus outperforms the RND.

Also, for ease of comparison, we provide the training curves of SAC-RND on three datasets: hopper-medium, halfcheetah-medium, and walker2d-medium in Figure 13.

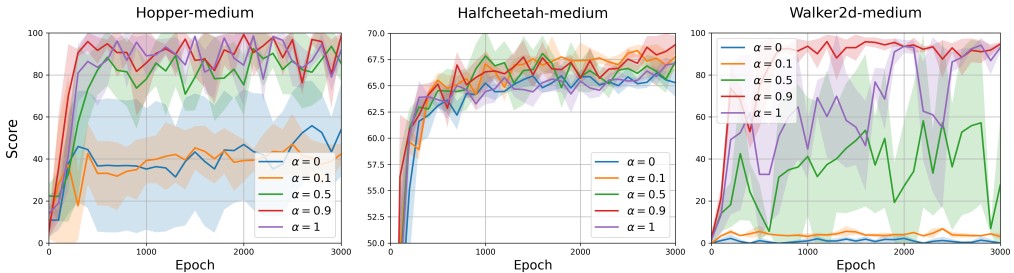

Figure 12: Training curves with different $\alpha$. All curves are averaged over 5 runs.

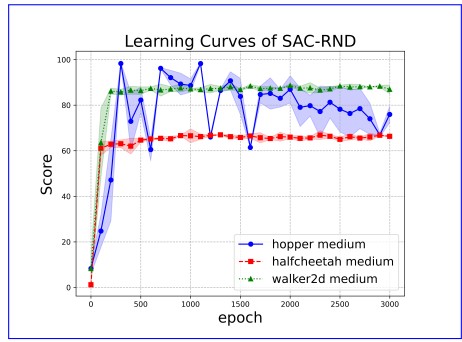

Figure 13: Learning curves of SAC-RND. The parameters are the same as in the original paper.

Table 9: The final scores of different $\alpha$ in offline tasks. We also compare the results of SAC-RND since it is a special case when $\alpha = 1$ and $N = 1$.

| Dataset $\qquad\qquad\alpha$ | 0.0 | 0.1 | 0.5 | 0.9 | 1.0 | SAC-RND |
|---|---|---|---|---|---|---|
| hopper-medium | 54.0 ± 31.1 | 42.3 ± 4.95 | 85.5 ± 8.7 | **98.5** ± 5.6 | 91.9 ± 4.9 | 91.1 ± 10.1 |
| halfcheetah-medium | 65.3 ± 1.3 | 67.5 ± 1.2 | 67.3 ± 0.6 | **68.6**± 0.4 | 67.1 ± 0.2 | 66.4 ± 1.4 |
| walker2d-medium | -0.0 ± 0.1 | 3.1 ± 1.7 | 27.6 ± 31.9 | **94.7** ± 1.0 | 93.0 ± 1.8 | 91.6 ± 2.8 |
| average score | 39.8 | 37.6 | 60.1 | **87.2** | 84.0 | 83.0 |

### E.7 EVALUATION ON OFFLINE-TO-ONLINE D4RL

We report offline-to-online performance on AntMaze tasks. We followed the methodology outlined by Tarasov et al. (2022). We report the scores after the offline stage and online tuning in Table 10.

| Task Name | TD3+BC | IQL | ReBRAC | SAC-DRND |
|---|---|---|---|---|
| antmaze-umaze | 66.8 → 91.4 | 77.00 → 96.50 | 97.8 → **99.8** | 95.8 → 98.3 |
| antmaze-umaze-diverse | 59.1 → 48.4 | 59.50 → 63.75 | 85.7 → **98.1** | 87.2 → 98.0 |
| antmaze-medium-play | 59.2 → 94.8 | 71.75 → 89.75 | 78.4 → 97.7 | 86.2 → **98.3** |
| antmaze-medium-diverse | 62.6 → 94.1 | 64.25 → 92.25 | 78.6 → **98.5** | 83.0 → 95.9 |
| antmaze-large-play | 21.5 → 0.1 | 38.5 → **64.50** | 47.0 → 39.5 | 53.2 → 51.5 |
| antmaze-large-diverse | 9.5 → 0.4 | 26.75 → 64.25 | 66.7 → **77.6** | 50.8 → 55.9 |
| **Average** | 46.4 → 54.8(+8.4) | 56.29 → 78.50(+22.21) | 75.7 → **85.2**(+8.5) | 76.0 → 83.0(+7.0) |

Table 10: Evaluation on Offline-to-online Setting. We compared the TD3+BC, IQL and ReBRAC algorithms, and their values were copied from Tarasov et al. (2023).

## E.8 MORE DETAILED CHANGE PROCESS OF RND BONUS

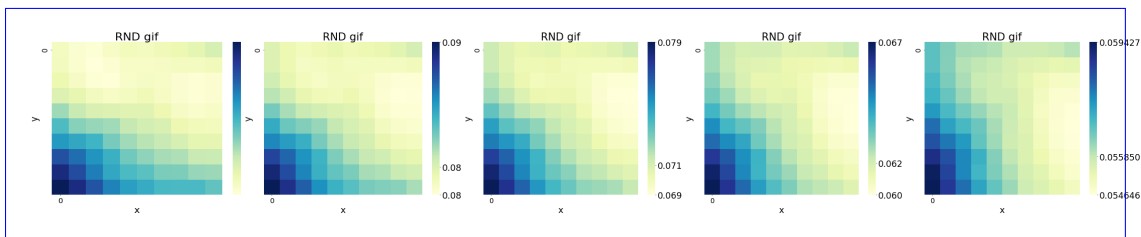

Figure 14: More Detailed Change Process of RND Bonus.

