# OpenReview forum: "Exploration and Anti-Exploration with Distributional Random Network Distillation"
_ICLR.cc/2024/Conference — Submitted to ICLR 2024_

### Official Review · Reviewer_4Wh1 · 2023-10-17

**Soundness:** 3 good
**Presentation:** 3 good
**Contribution:** 3 good
**Rating:** 6
**Confidence:** 3

**Summary:**

This work extends the Random Network Distillation (RND) algorithm and thus proposes Distributional RND (DRND). The DRND enhances the exploration process by distilling a distribution of random networks and implicitly incorporates pseudo counts to improve the precision of bonus allocation. This refinement encourages agents to engage in more extensive exploration. The proposed method excels in challenging online exploration scenarios and effectively serves as an anti-exploration mechanism in D4RL offline tasks.

**Strengths:**

1.	This paper is well-written and has good readability.
2.	The authors improve the RND in a simple but effective way.
3.	The proposed method get the high performance on offline tasks.

**Weaknesses:**

1.	I would like to see the learning curves of SAC-RND and SAC-DRND on the hopper, halfcheetah, and walker2d.
2.	$\lambda$ is carefully tuned for different datasets, therefore, the sensitivity analysis on this parameter should be given.
3. A similar work should be cited (Controlling Overestimation Bias with Truncated Mixture of Continuous
Distributional Quantile Critics).

**Questions:**

1.	How to test SAC on offline tasks?  Why the results of SAC in Table 1 are so low?
2.	In Fig.2 (b), there is no backpropagation from intrinsic reward. In this case, how to optimize the policy?
3.	What is the desired distribution in Fig.1?

---

> ### Author Response · Authors · 2023-11-18
>
> All minor issues will be addressed in the revision. Please check it.
> ### Weaknesses
> 1. We provide learning curves of SAC-RND on hopper-medium, halfcheetah-medium, and walker2d-medium in Appendix E.6.2 of revision, please check it in. Corresponding DRND curves can be found in Appendix E.4.2 in our original paper.
> 2. we have included sensitivity analysis results, please refer to Appendix E.3 in the revision.
> 3. Thanks for the suggestion. Reference has been added.
> ### Questions
> 1. Treating the offline dataset as the replay buffer for SAC. SAC results are taken from [_Uncertainty-Based Offline Reinforcement Learning with Diversified Q-Ensemble_](https://arxiv.org/abs/2110.01548). The score is low because SAC is a method for online settings. When learning directly from offline datasets, it will cause extrapolation errors[1][2]. Extrapolation errors will lead to serious overestimation of the value function and then lead to the collapse of policy learning.
> 2. The intrinsic rewards are added to the original rewards and the policy is updated using the original RL methods. The backpropagation in Fig.2 is for updating additional modules (RND and DRND), not for policy.
> 3. The bonus distribution should be inversely proportional to the dataset distribution. For the dataset shown in Fig.1(left), the desired distribution would be such that areas with darker colors in the dataset distribution have lighter colors in the bonus distribution, and areas with lighter colors in the dataset distribution have darker colors in the bonus distribution.
>
> [1] Off-Policy Deep Reinforcement Learning without Exploration
>
> [2] Stabilizing Off-Policy Q-Learning via Bootstrapping Error Reduction

---

> > ### Author Response · Authors · 2023-11-21
> >
> > Dear Reviewer u9ew,
> >
> > Thank you for your constructive review!
> >
> > May we kindly ask whether our response has addressed your concerns on the contribution of our work, the learning curves of SAC-RND, the sensitivity analysis, the reason of SAC scores, the way to optimize the policy?
> >
> > We will be more than happy to address any of your remaining/further questions during the discussion period. If our response has addressed your concerns, would you mind re-evaluating our work based on the updated information?

---

### Official Review · Reviewer_u9ew · 2023-10-24

**Soundness:** 2 fair
**Presentation:** 3 good
**Contribution:** 2 fair
**Rating:** 6
**Confidence:** 2

**Summary:**

Authors propose an improvement for the Random Network Distillation (RND) by employing N target networks instead of one. Additionally bonus calculation is splitted into two components. Authors claim that their approach DRND outperforms RND and other baselines in offline and online RL settings.

**Strengths:**

Authors show the problems that occur with RND approach and propose reasonable idea with mathematical proofs to overcome them. Empirical results could be be another strong side but, unfortunately, I have many concerns about them, see **Weaknesses**.

**Weaknesses:**

**Offline RL results**

Authors report baseline offline RL scores from ReBRAC paper but half of the scores do not match. For example, in case of ReBRAC MuJoCo scores are reported correctly for each dataset but average is changed to the lower value from 81.2 to 78.0. ReBRAC's scores on AntMaze match original paper except antmaze-large-diverse where they are swapped between ReBRAC and SAC-RND which leads to ReBRAC's average performance 75.4 instead of 76.8 which is better than DRND.

SAC-RND performs poorly on Adroit D4RL domain and DRND is not tested on it. DRND performs better only on MuJoCo tasks. I could admit experimental results to be strong if DRND's performance on Adroit is competitive to ReBRAC or IQL.

**Online RL results**

I haven't seen references to online RL baselines results. Problems with offline RL results make me question reported plots reliability. Please, correct me if I've missed it.

**Hyperparameters choice**

Hyperparameter choice procedure is also not clear from the text, please indicate if I've missed it. Hyperparameters candidates for $\lambda$ parameters are not presented and values are different for every offline dataset. Both, SAC-RND and ReBRAC, also tune this parameters for every dataset but they also report sensitivity to hyperparameters.
ReBRAC's authors tuned parameters on a few seeds and then evaluated all of the algorithms on another set of hyperparameters which lead to the SAC-RND performance drop comparing to the original work (SAC-RND reported results without evaluating on different set of seeds). Could you please say if the same procedure followed your work?

**Questions:**

Most of the questions are based on the **Weaknesses**. I understand that all of the experiments is hard to run during the short rebuttal phase but I kindly ask you to run at least some of them.

* Please fix scores for offline RL baselines or explain the change in scores.

* Could you provide reference scores for online baselines?

* What is DRND performance on Adroit tasks?

* What is hyperparameters choice procedure?

* What is hyperparameters sensitivity? It can be easily done with EOP (https://arxiv.org/abs/2110.04156) if there are logs from hyperparameters search. ReBRAC provide EOP scores for ReBRAC and IQL which can be reused for comparison.

* How DRND would perform in offline-to-online setting, e.g. on AntMaze? It is an open question for SAC-RND. Since DRND is reported to have good online performance it can be potentially used in offline-to-online setup. You can compare with the results from ReBRAC's latest revision.

---

> ### Author Response · Authors · 2023-11-18
>
> All minor issues will be addressed in the revision. Please check it.
> ### Questions:
> - We have checked all of the scores in our paper. The ReBRAC score in MuJoCo is 81.2, not 78.0. The SAC-RND score of antmaze-large-diverse is 45.7±28.5. The ReBRAC score of antmaze-large-diverse is 54.4±25.1 and its average is 76.8. We apologize for all mistakes in transcription and calculation. The other values are no problem.
> - The Atari scores can be found in the RND paper's Figure 7. For example, the last value of MontezumaRevenge is about 7500 and Gravitar is 4000. We run a PyTorch version of RND from [random-network-distillation-pytorch](https://github.com/jcwleo/random-network-distillation-pytorch). Its results are slightly lower than the original tensorflow version. The other results can be found in CFN paper, but the authors do not open their complete code. We replicated all the other baselines based on [CleanRL](https://github.com/vwxyzjn/cleanrl). To ensure that all results can be replicated, all of our codes are included in the supplementary material.
> - We conducted experiments with DRND in the Adroit environment, and the results are shown in the table below:
>
> | Dataset | DRND Score | $\lambda_{actor}$ | $\lambda_{critic}$ |
> | ------- | ---------- | -----------| ----------|
> | Pen-expert | 65.0±17.1 | 10.0| 5.0|
> | Pen-human | 42.3±11.8 | 5.0| 10.0|
> | Pen-cloned | 39.5±33.4 | 0.5 | 0.5 |
> | Door-expert | 85.3±37.9 | 10.0 | 10.0 |
> | Door-human | 1.3±0.8 | 5.0 | 0.01 |
> | Door-cloned | 0.3±0.1 | 5.0 | 5.0 |
> | Hammer-expert | 37.1±47.2 |2.5| 10.0|
> | Hammer-human | 0.3±0.2 | 10.0 | 0.01 |
> | Hammer-cloned | 1.1±0.8 | 1.0 | 1.0 |
> | Relocate-expert | 10.1±7.1 | 5.0 | 0.1 |
> | Relocate-human | 0.0±0.1 | 10.0 | 1.0 |
> | Relocate-cloned | 0.0±0.0 | 10.0 | 10.0 |
> | Average w/o expert | 10.6 | | |
> | Average | 23.5 | | |
> The final average score was not high. According to our experience, policy constraint methods often obtain better performance on Adroit. Some methods that do not include policy constraints, such as EDAC, often do not perform well on adroit. In addition, it is worth mentioning that RORL also introduced behavior cloning term in the AntMaze task to achieve a high score. Therefore, we believe that a feasible direction to obtain good results on adroit is to add explicit policy constraints to SAC-DRND. Furthermore, there is still a significant improvement compared to SAC-RND, from 12.9 to 23.5. This illustrates the superiority of DRND compared to RND. In addition, the average score without using the expert dataset has also improved significantly, reaching a level comparable to CQL(11.7), which benefits from the performance in the Pen environment.
> - The procedure of hyperparameter choice is as follows: we predefined a range of hyperparameters (actor alpha and critic alpha), and then conducted numerous experiments to search for the best hyperparameters. Specifically, we ran each pair of parameters for each task on 5 seeds and then evaluated the model online with 10 random seeds. The other parameters are the same as SAC-RND.
> - We supplemented the EOP of all experiments in the MuJoCo and AntMaze environments to analyze hyperparameters sensitivity. The results are shown below. For more specific curves, please refer to Appendix E.3 in the revision.
>
> |Domain|1 policy|2 policies|3 policies|5 policies|10 policies|15 policies|20 policies|
> |-----|-----|-----|-----|-----|-----|-----|-----|
> |Gym-MoJoCo|69.9±30.1|73.2±19.0|79.4±11.9|82.5±7.8|84.0±6.0|84.9±3.1|85.3±2.0|
> |AntMaze|69.3±15.9|75.3±10.1|78.5±7.6|81.5±4.0|83.7±3.1|84.5±1.5|84.9±0.9|
> - We are doing this experiment. If we can get the results before the end of the rebuttal, we will update it in the latest revision.
>
> The results show that SAC-DRND exhibits higher EOP than ReBRAC in MuJoCo and AntMaze. It can also be seen from the figures we showed in the revision that in most tasks, EOP converges within 10 to 20 evaluations.

---

> ### Author Response · Authors · 2023-11-21
>
> Dear Reviewer u9ew,
>
> Thank you for your constructive review!
>
> May we kindly ask whether our response has addressed your concerns on the contribution of our work, the score issue of ReBRAC, the online experiments, the DRND performance on Adroit tasks, hyperparameters choice procedure and hyperparameters sensitivity?
>
> We will be more than happy to address any of your remaining/further questions during the discussion period. If our response has addressed your concerns, would you mind re-evaluating our work based on the updated information?

---

> ### Comment · Reviewer_u9ew · 2023-11-21
>
> I thank authors for their rebuttal experiments. I think most of my concerns are resolved. Please also fix the scores in the appendix

---

> > ### Comment · Reviewer_u9ew · 2023-11-21
> >
> > I've raised my score but deceeased the confidence. Hope you understand that I can't be very sure about raising the score from 3 to 6.

---

> > > ### Author Response · Authors · 2023-11-21
> > > **Thank you**
> > >
> > > We thank the reviewer for raising the score to 6! We now can confirm that all of the numerically reported values (either main paper or the appendix) are correct in our latest revision. Thank you for the very helpful comments.

---

> > ### Author Response · Authors · 2023-11-21
> >
> > We apologize for the bugs in the appendix. Thanks for pointing it out. We have corrected the scores in the appendix. We wonder whether the reviewer is satisfied with our latest revision. We are fully prepared to do our best to address any remaining concerns raised by the reviewer. If most of the concerns are addressed, we hope the reviewer can re-consider the score.

---

### Official Review · Reviewer_gfXh · 2023-11-04

**Soundness:** 3 good
**Presentation:** 2 fair
**Contribution:** 3 good
**Rating:** 6
**Confidence:** 3

**Summary:**

This paper identifies that intrinsic reward bonuses from Random Network Distillation (RND) exhibit inconsistency with respect to the experienced data distribution. Initial bonuses from RND can be non-uniform, and final bonuses may not correspond to the distribution of state visitations.

To overcome this limitation, this paper proposes DRND, which samples one of N random target networks to generate targets for the predictor. The newly proposed intrinsic reward combines two bonuses, one which generalizes RND and the other one generalizes Coin Flip Networks (CFN).

Theoretical analysis in the linear setting supports the idea, and experiments in online and offline settings show that the proposed implementation can provide benefits over RND and other bonus-based approaches.

**Strengths:**

**S1.** RND is a popular approach to exploration with intrinsic rewards, and the identified limitation of RND and the proposed DRND approach would interest the research community.

**S2.** The proposed approach is interesting and novel (to the best of my knowledge).

**S3.** DRND is computationally cheaper than ensemble-based approaches, which could be used to resolve the bonus inconsistency. An advantage of using multiple target networks with only a single predictor is that it does not require maintaining an ensemble of predictors, which would require separate backward passes.

**S4.** Experiments have been conducted in a diverse range of environments.

**Weaknesses:**

**W1.** My main concern is with the empirical evaluation. Using state-actions for DRND and only states for RND is not an exact comparison; there should be analysis/ablations where DRND uses just states, and RND uses state-actions.

This might be particularly important as the performance difference between RND and DRND is small in many environments.

**W2.**  The proposed approach computes intrinsic rewards by combining two bonuses. However, I feel that introducing the second bonus–similar to the bonus from Coin Flip Network (CFN)--is not sufficiently well-motivated. Could the authors explain the need for the second bonus? Why is $b_1$ alone insufficient?

**W3.** The paper would benefit from comparisons with ensemble-based RND in the online setting, for example, along the lines of [1] or [2]. Even though ensemble-based approaches may be computationally more costly, having these results would be helpful data points.

**W4.** The presentation of the paper can be improved. Some suggestions are listed below.

- The intrinsic reward paragraph in the preliminaries is unclear. What is $y_t$ the target for?
- The RND method is defined with observations, but the preliminaries define an MDP with states.
- Some sentences are hard to understand, e.g., “and its interpretability needs to be more apparent than count-based techniques” and “These approaches are implemented through formulas such as $r_t =$ ..”
- What does the shading in figures represent?
- It would also be helpful to show DRND’s performance in Figure 1 rather than having Figures 1 and 3 separate.
- The term “deep exploration” should be explained in the paper or supported by citations.

Overall, I appreciate the direction the authors took toward improving RND-like intrinsic bonuses. I remain open to increasing the score should the weaknesses and questions be adequately addressed/clarified.

—------------------—------------------—------------------—------------------—------------------

### References

[1] Ciosek, K., Fortuin, V., Tomioka, R., Hofmann, K., & Turner, R. (2019). Conservative uncertainty estimation by fitting prior networks. In International Conference on Learning Representations.

[2] Ramesh, A., Kirsch, L., van Steenkiste, S., & Schmidhuber, J. (2022). Exploring through random curiosity with general value functions. Advances in Neural Information Processing Systems

**Questions:**

Q1. What were the architectures of random and target networks used in the experiments for Figures 1 and 3?

Q2. Is there evidence for bonus inconsistency in the tasks other than the one used for Figure 1? And if so, would DRND resolve it?

Q3. Coming back to the experiment in Figure 1, it would also be useful to see snapshots of the bonus inconsistency at more points through learning, and not just the beginning and the end. Are these results available?

---

> ### Author Response · Authors · 2023-11-18
>
> All minor issues will be addressed in the revision. Please check it.
> ### Weaknesses
> W1: We use state-actions as inputs for offline experiments(both DRND and RND) and only states for online experiments. Because in online experiments, RND/DRND is used for exploration. The purpose of the module is to determine what states are rarer and more worth exploring for agent. The calculation of action is not involved. However, in offline setting, RND/DRND is for anti-exploration, specifically, the module is for finding what state-action pairs are in dataset distribution and what not. So the different input type depends on task setting, not RND or DRND module.
>
> W2: Sorry for confusion. The two bonuses come from the two bonus inconsistencies we found in RND, please refer to Section 4.1. Without the second term, the second inconsistency will not be resolved because we have shown that the second term bonus is an unbiased estimate of 1/n. This difference of two bonuses is shown in Fig.3 (middle and right). The first bonus flattens the initial bonus distribution. The second bonus aligns the final bonus distribution with the data distribution. In numerical terms, the initial value of the first term is relatively large, but with gradient backpropagation, the second term gradually becomes dominant.
>
> W3: Thanks for your suggestions and we compared ensemble-based RND. We are doing this experiment. If we can get the results before the end of the rebuttal, we will update it in the latest revision.
>
> W4:
> - $y_t$ is the target of Q-network.
> - Appology for confusion. We will unify to observation in revision.
> - We will check the whole paper and refine this. Thanks.
> - Shading indicates standard deviation intervals.
> - Thanks for suggestion. Will refine it.
> - We have added citation in revision, please check it.
> ### Questions
> Q1: The architecture of predictor networks is `nn.Sequential(nn.Linear(2, 16), nn.ReLU(), nn.Linear(16, 16), nn.ReLU(), nn.Linear(16, 16))`. Target networks' architecture is `nn.Sequential(nn.Linear(2,16),nn.ReLU(), nn.Linear(16, 16))`.
>
> Q2: We did another set of experiments different from Figure 1 in the paper, please refer to Section 5.1. Bonus inconsistency also appears, and DRND solves it better. We analyzed the impact of different target numbers on bonus inconsistency in it.
>
> Q3: The results are available. We have turned more data into revision, please refer to Figure 1 and Appendix E.7 in the lastest version.

---

> > ### Author Response · Authors · 2023-11-21
> >
> > Dear Reviewer gfXh,
> >
> > Thank you for your constructive review!
> >
> > May we kindly ask whether our response has addressed your concerns on the contribution of our work, the input of both DRND and RND, the role of the second bonus?
> >
> > We will be more than happy to address any of your remaining/further questions during the discussion period. If our response has addressed your concerns, would you mind re-evaluating our work based on the updated information?

---

> > > ### Comment · Reviewer_gfXh · 2023-11-21
> > >
> > > Thank you for your response. I appreciate the time and effort spent in revising the previous version. Some of my questions still remain open.
> > >
> > > Regarding W1, thanks for the clarification, but this is not clear from the paper. At the beginning of section 4.2, we have
> > >
> > >
> > >  >“ Unlike RND, which only has one target network f (s), the DRND algorithm has N target networks f ̄(s,a ),f ̄(s,a) ,..., f ̄ (s,a) …….  and do not participate in training. In DRND, we have also considered action, which can increase the exploration of action in the same state.”
> > >
> > > This indicates that RND uses only states (f(s)) and DRND uses state-action pairs. Could the authors please confirm if the paper is accurate. Further, to respond to your comment that only state novelty is relevant for exploration, I think looking at novelty in state-action space is not unreasonable.
> > >
> > > Regarding W2, the motivation for combining the two bonuses is still unclear to me. I see the results in Figure 3, and I think both bonuses are sensible in their own right. But why are we combining them? My question is coming from a conceptual point rather than the empirical motivation of Figure 3. Wouldn’t the first bonus also align with the data distribution (as it eventually converges)? Similarly, would the authors argue that combining RND and CFN bonuses is a sensible idea?
> > >
> > > Regarding W4, is the shading one standard deviation or two (for a 95% interval)?

---

> > > > ### Author Response · Authors · 2023-11-21
> > > >
> > > > For W1, what we said in the rebuttal is a bit of an overclaim. Using state-action in an online setting is also fine. Initially, we intended to replace both RND and DRND in the online and offline settings with state-action pairs.   However, all RND baselines only consider the state by default. Following them, our DRND algorithm also considers only the state. In our offline experiments, our inputs consisted of both state and action, facilitating alignment with other offline algorithms. We sincerely apologize for the inaccuracies in the article. We will rectify the representation in the revision.
> > > >
> > > > For W2, we agree that both bonuses are sensible in their own right. However, they are not enough when working alone. From a conceptual standpoint, the first bonus is the MSE loss between predictor and targets. We believed that using this bonus alone as an intrinsic reward might lose its distinctiveness after numerous updates, e.g., the loss after 100 updates is fairly close to the loss after 1000 updates as the loss function converges. Consequently, in environments requiring hard exploration, this approach might not work. Hence, we introduce the second bonus as a complementary, devising a statistical estimate for access count. As shown in the paper, theoretically, the estimation is inaccurate with limited data, becoming precise only after gathering a substantial amount of data. These two aspects complement each other directly, naturally leading to our design choice that includes both components. We chose the simplest method of weighted summation as a bonus for experimentation and found that the combined effect was quite promising. The experiments in Appendix E.6 have shown that the first term does not fully align with the data distribution (even if it eventually converges). In summary, we would argue that combining RND and CFN bonuses is a sensible idea.
> > > >
> > > > For W4, the shaded area is a standard deviation.
> > > >
> > > > We are happy to engage in further discussions if any remaining concerns persist.

---

> > > > > ### Comment · Reviewer_gfXh · 2023-11-21
> > > > >
> > > > > Thanks for clarifying and continuing the discussion. The paper would benefit from a careful rewrite to correct typos and inconsistencies. Based on your answer, I better understand your motivation behind having two bonuses now.
> > > > >
> > > > > I have some questions about the empirical evaluation (W1).
> > > > >
> > > > > Figure 5 (Atari): Is there a reason results are presented over episodes rather than steps/frames as is standard in these tasks?
> > > > > Furthermore, the results in Montezuma's revenge don't match the results presented in CFN (https://arxiv.org/pdf/2306.03186.pdf, figure 7), where CFN performs similarly to RND. Could the authors explain this mismatch?
> > > > >
> > > > > Figure 6 (Adroit): Similar to the above, the results don't match the CFN paper's results. In Door, CFN and RND are already close to 1 at 0.5 million steps (https://arxiv.org/pdf/2306.03186.pdf, figure 5), whereas they are much slower here. Also, see Relocate, where CFN from the original paper is at about 0.3 at 2 million steps, whereas in this paper, CFN reaches that point at around 5 million steps. I am also concerned with some results from Figure 7 (Fetch). Could the authors clarify why there are mismatches in CFN/RND performance?

---

> > > > > > ### Author Response · Authors · 2023-11-23
> > > > > >
> > > > > > Dear Reviewer gfXh,
> > > > > >
> > > > > > We appreciate your continued attention to the paper's details and value your concerns.
> > > > > >
> > > > > > May we kindly ask if our responses have addressed your questions about the empirical evaluation, the discrepancy in CFN performance between our paper and the original paper, as well as the variance in CFN performance and RND's?
> > > > > >
> > > > > > We would be happy to address any remaining or additional queries you may have.
> > > > > >
> > > > > > If our responses have resolved your concerns, would you consider re-evaluating our work based on the updated information?

---

> > > > > > > ### Comment · Reviewer_gfXh · 2023-11-23
> > > > > > >
> > > > > > > Thank you for the clarifications and continuing the discussion, I have updated my score accordingly. One remaining concern about this issue is that since the paper doesn't use the baseline exactly as is, more discussion is needed on how hyperparameters related to CFN were selected (please correct me if I missed this). After the discussions, I still feel the paper misses details like this which are important to convey.

---

> > > > > > > > ### Author Response · Authors · 2023-11-23
> > > > > > > >
> > > > > > > > We express our gratitude to the reviewer for the increased score!  We further supplemented and refined our experimental details in the appendix, please check Appendix C. In our experiments, the fundamental parameters of the base algorithm PPO such as learning rate and batch size were kept identical across all methods. For the hyperparameters of the utilized exploration algorithms, we utilized the author-recommended hyperparameters from respective papers (e.g., CFN). Specially, the hyperparameter $d$ specific to CFN corresponds to the output dimension of the predictor network, for which we set the output dimension as 32 in the appendix.

---

> ### Author Response · Authors · 2023-11-22
>
> For Figure 5 (Atari), we adopted a log method over the episode as we do not observe much difference in steps/frames number per episode across different baselines. We can update the figure with steps/frames as the x-axis if the reviewer thinks it necessary. On average, each episode contains 1200 steps. We used a total of 128 workers to collect data, so in Figure 5 (Atari), 2000 episodes correspond to about 128 * 1200 * 2000=300M steps (We use MonteZumaNoFrameSkip-v4, so frame is equal to step). 200M steps correspond to episode 1333 in Figure 5, and the score of RND is about 5000, which is the same as the result of RND in CFN (https://arxiv.org/pdf/2306.03186.pdf, Figure 7, the score of RND at 200M is 5000).
>
> The following reasons may cause the inconsistency: we use the PPO algorithm, which is widely adopted in online experiments, as the base algorithm instead of the SAC used in the CFN paper (please check CFN paper Section 4 **Implementation details**). We selected the same basic algorithm for all methods for a fair comparison. The difference between PPO and SAC leads to a mismatch between our reported performance and the results presented in the CFN paper. The sample efficiency of PPO is much worse than that of SAC (https://arxiv.org/abs/1802.09477, figure 5).
>
> The reason why CFN is worse than RND: some tricks are used in the CFN paper(https://arxiv.org/pdf/2306.03186.pdf, Section 3.5 **Improving predictions for novel states**), which are no longer applicable to the on-policy algorithm like PPO (the prioritized sampling trick and optimistic initialization trick cannot be used as PPO does not have a replay buffer). This may make our results different from the reported ones in the original paper. Meanwhile, the original code of CFN uses TensorFlow and is not fully disclosed. We reproduce CFN with PyTorch. These we believe all can contribute to the different reproduction results. In addition, we provide all code and ensure that all results are reproducible.

---

### Author Response · Authors · 2023-11-18
**Summary of Revision**

We submitted a revision. All changes are marked in blue font or boxes. Here are the changes:
- Added reference [1] of the term deep exploration into the introduction.
- Simplified some semantically challenging sentences (highlighted in blue font).
- Added reference into [2] in Related Work.
- The definition of MDP is unified as observation in Preliminaries; the meaning of $y_t$ is explained in the part of Intrinsic Reward.
- Added the network structure of the experiment in Figure 1 and Figure 3 (Appendix C).
- Added the change process of Figure 1 (Appendix E.8).
- Added explanation of shaded parts of images (Section 5.2).
- Modified the expression of the input of DRND in section 4.2.
- Modified the incorrect score in the table about SAC-RND (antmaze-large-diverse's score) and ReBARC (Gym-MoJuCo's average, antmaze-large-diverse's score, and AntMaze's average) in Section 5.3 and Appendix E.1, as well as the corresponding expressions.
- Supplemented the experiment of SAC-DRND on Adroit tasks (Appendix E.2).
- Supplemented the experiment of hyperparameters sensitivity (Appendix E.3). We provide EOP of SAC-DRND and all EOP curves on Gym-MuJoCo and AntMaze.
- Added learning curves of SAC-RND for simpler comparison (Appendix E.6.2).
- Added offline-to-online scores on AntMaze tasks (Appendix E.7).

In addition, to meet the page count requirement, we have deleted part of Section 6.

[1] Deep exploration via bootstrapped dqn

[2] Controlling overestimation bias with truncated mixture of continuous distributional quantile critics

---

### Meta-Review · Area_Chair_cGJX · 2023-12-06

**Metareview:**

This paper introduces a simple theoretically-grounded extension of the popular RND method by measuring novelty with respect to the output of randomly distributed random network, rather than a single random network. The authors mathematically argue that this approach addresses bonus inconsistency issues that afflict standard RND and show that this method combines aspects of both RND-style novelty estimation and exploration based on pseudocounts. The experiments show that this new approach leads to significant gains on continuous control tasks.

The paper is clearly written and the method is well-motivated. However, the weakness in this work is primarily around the experiments validating the performance of their method and its underlying design choices. Notably, while the intrinsic bonus for DRND is defined as the weighted average of two terms, according to some weight $\alpha < 1$, there are no comparisons of how the choice of $alpha$ impacts performance. Such comparisons would also serve as ablations for the importance of each term in the intrinsic bonus—an important set of results to justify the method's design.

The paper also studies exploration solely in the context of continuous control tasks, many of which are not designed as hard exploration problems. DRND's performance on many of these tasks appear to be within noise of RND's performance. There are many benchmarks specifically designed for testing exploration in RL, such as Montezuma's Revenge, on which the authors are encouraged to evaluate DRND with respect to baseline methods in future versions of this work.

**Justification For Why Not Higher Score:**

The paper is recommended for rejection due to the weaknesses outlined in the meta-review. While the high-level ideas behind the method are well-motivated, the experiments do not fully justify the design choices and effectiveness of the method as an improvement over RND in hard-exploration domains.

**Justification For Why Not Lower Score:**

N/A

---

### Decision · Program_Chairs · 2024-01-16

Reject